# Uncovering disease-related multicellular pathway modules on large-scale single-cell transcriptomes with scPAFA

Zhuoli Huang [1,2], Yuhui Zheng [1,2], Weikai Wang[1,2], Wenwen Zhou[1,2], Yanbo Zhang[3,4], Chen Wei[1,2], Xiuqing Zhang[1,2], Xin Jin [1,2,3] ✉ & Jianhua Yin [2,3] ✉

Pathway analysis is a crucial analytical phase in disease research on single-cell RNA sequencing (scRNA-seq) data, offering biological interpretations based on prior knowledge. However, currently available tools for generating cell-level pathway activity scores (PAS) exhibit computational inefficacy in large-scale scRNA-seq datasets. Additionally, disease-related pathways are often identified through cross-condition comparisons within specific cell types, overlooking potential patterns that involve multiple cell types. Here, we present single-cell pathway activity factor analysis (scPAFA), a Python library designed for large-scale single-cell datasets allowing rapid PAS computation and uncovering biologically interpretable disease-related multicellular pathway modules, which are low-dimensional representations of disease-related PAS alterations in multiple cell types. Application on colorectal cancer (CRC) datasets and large-scale lupus atlas over 1.2 million cells demonstrated that scPAFA can achieve over 40-fold reductions in the runtime of PAS computation and further identified reliable and interpretable multicellular pathway modules that capture the heterogeneity of CRC and transcriptional abnormalities in lupus patients, respectively. Overall, scPAFA presents a valuable addition to existing research tools in disease research, with the potential to reveal complex disease mechanisms and support biomarker discovery at the pathway level.

Single-cell RNA sequencing (scRNA-seq) technologies can facilitate the high-throughput and high-resolution profiling of single-cell level transcriptomes, which have revolutionized molecular biology[1]. In the past few years, the prevalence of scRNA-seq and the implementation of sample multiplexing technique[2] have led to the emergence of large-scale single-cell transcriptome atlas. Representative examples of large-scale atlases containing peripheral blood mononuclear cell (PBMC) atlas with over 1.2 million cells from 99 healthy controls and 162 systemic lupus erythematosus (SLE) cases[3]; COVID-19 comprehensive immune landscape with 1.46 million cells from 196 individuals[4]; lung atlas with over 2.4 million cells from 486 individuals[5]; adult human brain dataset comprised more than 3 million cells[6]. These atlases function as indispensable resources for delving into the intricacies of both healthy and diseased cellular states; nevertheless, they concurrently place higher requirements on the stability and efficiency of analytical methods.

Pathway analysis constitutes a pivotal analytical phase in the interpretation of omics data, facilitating the detection of alterations in cellular biological processes. Existing single-cell pathway activity scoring methods, such as AddModuleScore[7,8], UCell[9], and AUCell[10] can generate pathway activity scores (PAS) for individual cells, rather than cell populations or clusters, and these scores can then be used for downstream analysis[11], such as clustering, cell type identification and inter-group comparisons based on biological conditions (e.g., case-control comparisons). Most scRNA-seq studies evaluate pathway activation centered on curated gene lists compiled by domain experts, which represent the current reference biological knowledge[7]. In contrast to utilizing the expression levels of individual genes, pooling of gene set-based measurements, which amalgamate the functional effects of diverse genes participating in identical biological pathways, can significantly augment statistical robustness and facilitate biological interpretation when discerning specific cellular functions or states, especially in sparse and noisy scRNA-seq dataset[12–14]. Nonetheless, the abovementioned methods exhibit tardiness and computational inefficacy in large-scale scRNA-seq datasets, especially in situations where a large number of pathways ($>10^3$) is required to be calculated. It is worth mentioning that a

[1]College of Life Sciences, University of Chinese Academy of Sciences, Beijing, 100049, China. [2]BGI Research, Shenzhen, 518083, China. [3]Shanxi Medical University-BGI Collaborative Center for Future Medicine, Shanxi Medical University, Taiyuan, 030001, China. [4]Department of Health Statistics, School of Public Health, Shanxi Medical University, Taiyuan, 030001, China. ✉e-mail: jinxin@genomics.cn; yinjianhua@genomics.cn

recently proposed sensitive method Single Cell Pathway Analysis (SCPA)[11] can deal with a large number of pathways, this approach directly provides a statistic (Qval) indicating the change of pathway activity in different conditions, instead of generating cell-level PAS. However, it uses a down-sampling strategy (selecting 500 cells per condition as default) that may lose information on large datasets, whereas increasing the number of down-sampled cells significantly affects computational efficiency.

A typical primary step in the analysis of scRNA-seq data is to partition the cells into clusters and annotate them as cell types[15]; therefore, the downstream analysis commonly prioritizes pairwise cross-condition comparisons centered around each interested cell type, neglecting the multicellular nature of disease processes[16]. For example, SCPA detects changes in pathway activity under different conditions in each cell type separately. We assumed that aggregating cross-condition differences of pathway activity scores detected in various cell types into a low-dimensional module would enable a more comprehensive elucidation and interpretation of the transcriptomic features representing the disease state. It was reported that multi-omics factor analysis (MOFA) can integrate data from complex experimental designs, including multi-omics data and various sample sets, such as experimental conditions or batch[17,18]. Gaining advantages from the high flexibility of the statistical framework, MOFA was widely applied in different biological contexts, for instance, revealing an axis of heterogeneity associated with the disease outcome in chronic lymphocytic leukemia[19]; identification of alterations in Alzheimer's disease[20]; uncovering multi-omics factors associated with adenoma and colorectal cancer risk[21]. Inspired by Ramirez et al.[16], we reutilized MOFA to identify multicellular pathway modules.

Here, we present single-cell pathway activity factor analysis (scPAFA), an open-source Python library designed for large-scale single-cell datasets to rapidly compute PAS and uncover disease-related MOFA-based multicellular pathway modules. The scPAFA contains time-efficient implementations of single-cell pathway activity scoring algorithms, which can compute PAS for 1383 pathways in million-cell-level scRNA-seq data within 30 min. It also provides user-friendly functions for PAS matrices preprocessing according to different experimental designs, MOFA model training, and downstream analysis of multicellular pathway modules. As a case study, we benchmarked scPAFA's performance on colorectal cancer (CRC) datasets[22,23] together with a large-scale lupus atlas[3]. We have demonstrated that scPAFA found trustworthy and interpretable multicellular pathway modules that represent the heterogeneity of CRC and transcriptional abnormalities in patients with SLE, respectively. Furthermore, high-weight features in the modules demonstrate outstanding performance in machine learning as input for classifier training.

## Results
### Overview of scPAFA workflow
The scPAFA workflow consists of four steps (Fig. 1a and Methods), each step of the scPAFA workflow is supported by user-friendly application programming interfaces (API) allowing customized parameters. In the first step, a single-cell gene expression matrix and collection of pathways are used to compute PAS by "fast_ucell" or "fast_score_genes". These functions are more computationally efficient implementations of UCell and AddModuleScore (also known as "score_genes" in Scanpy[24]), which are commonly used single-cell pathway activity scoring methods. In brief, the "fast_ucell" function reimplements the R package UCell in Python language, utilizing more vectorized computations and designing an efficient chunking and concurrent computation process to fully use multi-core CPUs; whereas the "fast_score_genes" is a concurrent implementation of the single-core "score_genes" function in Scanpy, designed with chunking capabilities. During the PAS computation, large datasets are first divided into multiple chunks, with each chunk containing 100,000 cells as default. For each pathway, the PAS calculation on a chunk is a fast, vectorized process supported by SciPy and NumPy; to deal with a large number of pathways, the pathways set is partitioned and distributed across multiple cores for parallel computation. This design enables scPAFA to quickly and efficiently convert

a cell-gene expression matrix into a cell-pathway PAS matrix ("Methods"). Collection of pathways can be obtained from public databases, such as the Molecular Signatures Database (MsigDB)[25,26] and National Center for Advancing Translational Sciences (NCATS) BioPlanet[27], and can be further customized based on specific biological contexts of interest.

Second, the single-cell PAS matrix is reformatted into a suitable input for the MOFA model, accompanied by cell-level metadata including sample/donor information, cell type, and technical batch details (if available). In brief, we reutilized the MOFA framework by assigning pathways as features, cell types as non-overlapping sets of modalities (also called views), and technical batch information (if available) as groups. Features will be centered per group before fitting the model to reveal which sources of variability are shared between the different groups, therefore batch effects are mitigated. If no significant batch effects are present, a single-group MOFA model can be applied. In each group and view, cell-level PAS is aggregated into pseudobulk-level PAS across samples/donors by computing the arithmetic mean. Consequently, the input to the MOFA model comprises pseudobulk-level PAS rather than cell-level PAS, enabling the model to be trained efficiently, typically within a few seconds.

In the third step, the MOFA model is trained via the "run_mofapy2" function of scPAFA, then the converged model is output. The latent factor matrices are extracted from the converged model using the "get_factors" function. These matrices are then scaled per factor per group, and subsequently integrated into a single latent factor matrix, with rows representing pseudobulk samples and columns representing factors. Similarly, the feature weight matrices are extracted using the "get_weights" function, scaled per factor per cell type (view), and integrated into a single weight matrix, with rows representing pathway-cell type pairs and columns representing factors. The factors integrate the primary axes of variation in PAS across conditions from different cell types and are considered as multicellular pathway modules, which can be interpreted by high-weight pathway-cell type pairs in the corresponding weight matrix. Finally, along with the sample-level clinical metadata, disease-related multicellular pathway modules (latent factors) can be identified by statistical analysis (Methods). Downstream analyses include characterizing and interpreting multicellular pathway modules, samples/donors stratification, and classifier training based on high-weight pathways.

### Validating computational efficiency of scPAFA
To demonstrate scPAFA's efficiency in computing PAS, we benchmarked it on two public scRNA-seq datasets in different diseases: colorectal cancer (CRC) dataset with 371,223 cells, collected from colorectal tumors and adjacent normal tissues of 28 mismatch repair-proficient (MMRp) and 34 mismatch repair-deficient (MMRd) individuals[22]; lupus dataset with 1,263,676 cells, collected from PBMCs of 162 SLE cases and 99 healthy controls[3]. We used NCATS BioPlanet[27], a single collection of known biological pathways operating in human cells which incorporates 1658 pathways as input. Specifically, for the CRC dataset, we additionally included 149 gene sets mined from the Curated Cancer Cell Atlas (3CA) metaprogram[28]. After quality control, 1629 and 1383 pathways were taken as input for the CRC dataset and lupus dataset, respectively. We first examined the PAS computation speed of scPAFA on an Intel X79 Linux server using 10 cores, comparing with UCell (10 cores), AUCell (10 cores), and "score_genes" (1 core) functions. It was observed that the running speeds of functions "fast_ucell" and "fast_score_genes" were ~3.8–47.4 times faster than their original versions on the dataset ranging from 10,000 cells to 1,263,676 cells. Compared to the widely used AUCell method, "fast_ucell" and "fast_score_genes" were ~4.4–11.4 times faster. For instance, on the complete lupus dataset, UCell costs 21.4 h for 1383 pathways, "score_genes" costs 9.3 h, AUcell costs 5.1 h; whereas "fast_ucell" costs 27.05 min, and "fast_score_genes" costs 29.9 min (Fig. 1b, Supplementary Data 1). The scPAFA can achieve greater computational efficiency gains on large-scale datasets for pathway activity scoring than existing methods. In addition, the peak memory usage of "fast_ucell" was lower than UCell (Fig. 1b). However, the memory usage of "fast_score_genes" was higher than "score_genes", due to

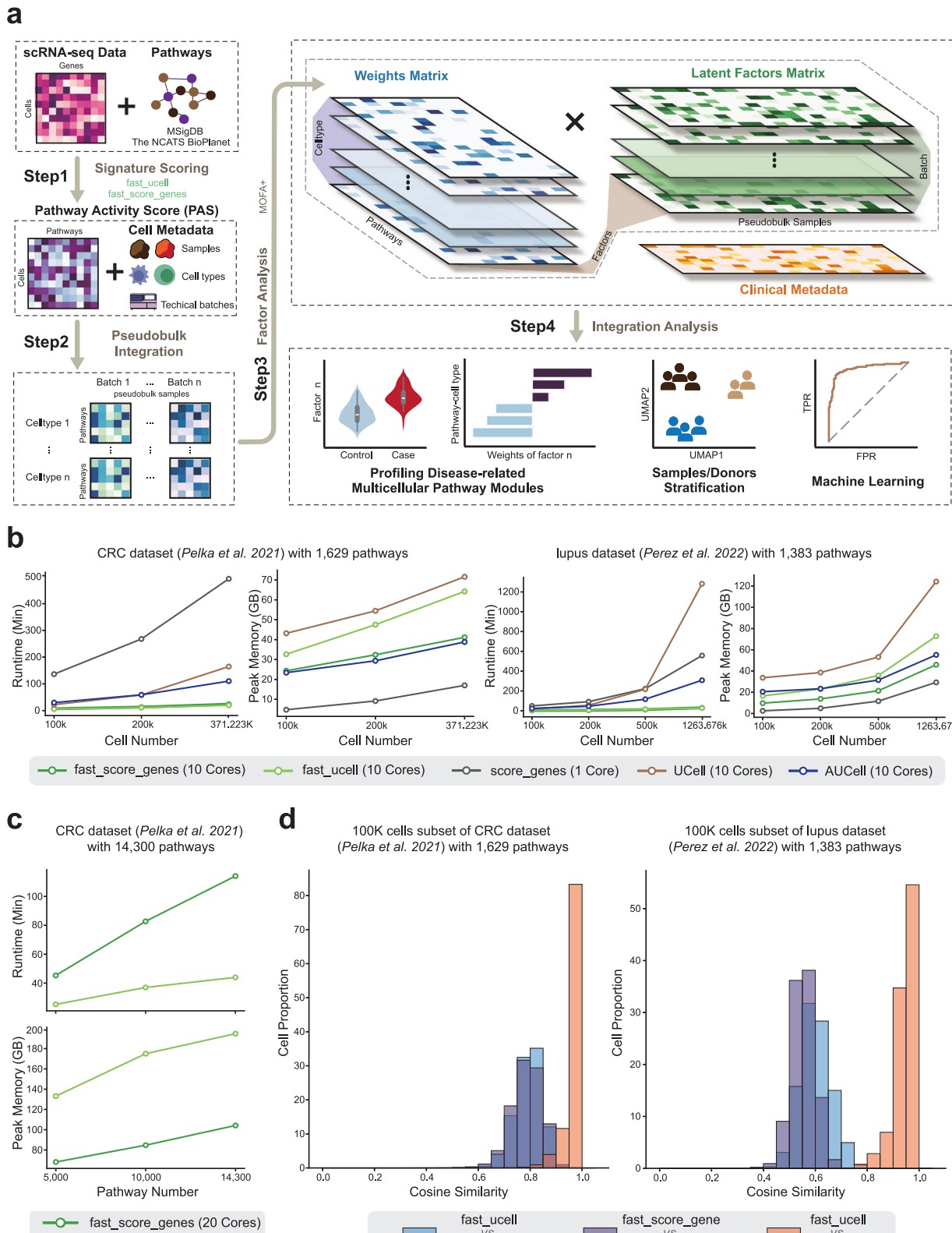

**Fig. 1 | scPAFA overview and its performance on CRC and lupus dataset.**
**a** Schematic outline of scPAFA workflow consists of four steps: 1) Generating PAS matrix from single-cell gene expression matrix and pathway set using "fast_ucell" or "fast_score_genes" functions. 2) Aggregating cell-level PAS matrix into pseudobulk-level PAS long data integrating with sample, cell type, and batch information. 3) Training MOFA model based on experimental design 4) Using latent factor matrix and its corresponding weight matrix of MOFA model to identify disease-related multicellular pathway modules and provide biological interpretation. **b** Line graphs show the runtime and the peak

memory usage of "fast_score_genes"(scPAFA, 10 cores), "fast_ucell"(scPAFA, 10 cores), "score_genes"(Scanpy, 1 core), UCell(R, 10 cores) and AUCell(R, 10 cores) on CRC scRNA-seq dataset (1629 pathways) and the lupus scRNA-seq dataset (1383 pathways). **c** Line graphs show the runtime and memory usage of "fast_score_genes"(scPAFA, 20 cores) and "fast_ucell"(scPAFA, 20 cores) on CRC scRNA-seq dataset (5000–14300 pathways). **d** Histogram showing the distribution of cell-level cosine similarities of PAS generated by "fast_score_genes", "fast_ucell", and AUCell in the CRC (100k cells subset with 1629 pathways) and lupus datasets (100k cells subset with 1383 pathways).

the memory overhead of concurrent execution, whereas the original "score_genes" function can only utilize a single core (Fig. 1b). Next, we tested the ability of scPAFA to handle large-scale pathway sets (>10⁴). We used C5 ontology gene sets from MsigDB along with the CRC dataset. After quality control, 14,300 pathways were included for PAS calculation. Using 20 cores, We found that "fast_ucell" costs 43.96 min and "fast_score_genes" costs 114.05 min, indicating 'fast_ucell' is more efficient in this context, whereas "fast_score_genes" cost less memory (Fig. 1c, Supplementary Data 1).

Moreover, UCell calculates PAS based on the Mann-Whitney U statistic[9], and AUCell uses the area under the curve (AUC) of the recovery curve. They both rely on the relative ranking of genes within individual cells, making their PAS robust to variations in dataset composition. We found that "fast_ucell" PAS was closely similar to AUCell PAS by setting the same max rank threshold, the average cosine similarity between 'fast_ucell' PAS and AUCell PAS was 0.97 and 0.94 in the CRC dataset and lupus dataset, respectively (Fig. 1d, Supplementary Data 1). Furthermore, "score_genes" normalizes its PAS against the average expression of a control set of genes across the entire dataset, making it sensitive to the composition. Consistently, the PAS of "fast_score_genes" has a moderate degree of similarity with the "fast_ucell" PAS or AUCell PAS (Fig. 1d, Supplementary Data 1). Considering "fast_ucell" can generate consistent PAS regardless of dataset composition, we use the PAS of "fast_ucell" for downstream input in this study.

## Multicellular pathway modules in tumor cells that stratify CRC patients

CRC is one of the most common cancers in developed countries. Immune responses to CRC are highly variable, tumors with a mismatch repair-deficiency (MMRd)/microsatellite instability-high (MSI-H) phenotype shows unique characteristics, including elevated tumor mutational burden and enhanced anti-tumor immunity than mismatch repair-proficient (MMRp) tumors[22,29]. Based on single-cell data and cell type annotations from the abovementioned CRC dataset[22], we further used scPAFA to identify MMR status-related multicellular pathway modules. In brief, we extracted the 108,497 epithelial tumor cells from the CRC dataset, which were reported as 11 cell clusters. The PAS matrix with 1629 pathways was computed by "fast_ucell", then aggregated along with samples and cell types information. We first used a single-group MOFA framework and revealed an obvious batch effect in samples' stratification caused by different 10X chemistry versions, which corresponded to the protocol for generating single-cell gene expression libraries (Supplementary Fig. 1a, b); hence PAS matrix was reformatted with batch, samples and cell type information while multi-group MOFA+ framework was further applied. The converged model provided a latent factor matrix with 8 factors that explain at least 1% of variance from 65 pseudobulk samples, the Uniform Manifold Approximation and Projection (UMAP) were used to perform dimensionality reduction and visualization on the latent factor matrix, showing distinct stratification between MMRd and MMRp CRC patients (Fig. 2a). The values of factor 2 and factor 3 were significantly higher in MMRd samples, whereas factor 6 demonstrated opposite trends (Fig. 2a). Noteworthy, pseudobulk samples from different batches exhibited non-segregated distribution on the UMAP plot, and the factor values showed no significant differences between batches, suggesting that no batch-related factors were detected (Fig. 2b). Moreover, two-dimensional scatter plot based on the combination of factors 2, 3, and 6 also showed stratification of pseudobulk samples by MMR status (Fig. 2c). Simultaneously, hierarchical clustering based on the values of factors 2, 3, and 6 also revealed MMR status-related stratification, additionally, heterogeneity among samples in each MMR status was also detected (Fig. 2d). Factors 2, 3, and 6 explained a higher proportion of variance in stemTA-like cell clusters (cE01, cE02, and cE03) compared to other cell types (Fig. 2e). Besides, factor 2 positively correlates with factor 3, while factor 6 negatively correlates with both factor 2 and factor 3 (Fig. 2f). In conclusion, the above results suggest that factors 2, 3, and 6 can be considered as tumor cell-based MMR status-related multicellular pathway modules.

## Pathway-centric interpretation of MMR status-related modules

We further employed the feature weights matrixes corresponding to factors 2, 3, and 6 to interpret MMR status-related multicellular pathway modules. Weights provide a score indicating the strength of the relationship between each feature and factor, for instance, pathway-cell type pairs with no association with the factor have values close to zero, while pathway-cell type pairs with stronger association have larger absolute values. The sign of the weight indicates the direction of the effect, consistent with the direction of the factor values. MMRd-related multicellular pathway modules consist of "Eosinophils in the chemokine network of allergy" in immature goblet, "METAPROGRAM_17_INTERFERON_MHC_II_1" in stemTA-like immature goblet and "METAPROGRAM_6_HYPOXIA" in stemTA-like immature goblet, etc., whereas MMRp related multicellular pathway modules included "METAPROGRAM_B_CELLS_METABOLISM_MYC" in stemTA-like cells, "Cap-dependent translation initiation" in enterocyte 2 and "Nef-mediated downregulation of MHC class I complex cell surface expression" in stemTA-like immature goblet, etc. (Fig. 3a, Supplementary Data 2). Furthermore, we validate the PAS differences between MMRd and MMRp of some high-weight pathway-cell type pairs extracted from feature weights matrixes corresponding to factors 2,3 and 6. We revealed that "METAPROGRAM_17_INTERFERON_MHC_II_1", a high-weight pathway associated with factors 2 and 3, had a significantly increased PAS in MMRd patients across various cell types compared with MMRp patients (Fig. 3b). Pelka et al.[22] reported that ISG and MHC class II gene programs were more active in MMRd versus MMRp tumors, which was consistent with our results. It is widely reported that MMRd CRC patients are more sensitive in response to anti-programmed death-1 receptor (PD-1)/programmed death-1 receptor ligand 1 (PD-L1) therapy than MMRp patients[30–32], consistently, PAS of 'PD-1 signaling' pathway in the immature goblet that associated with factor 2 was significantly elevated in MMRd patients (Fig. 3b). Moreover, PAS of factor 6 associated-pathway "METAPROGRAM_6_HYPOXIA" was significantly up-regulated in MMRd patients, which was consistent with previous research reporting hypoxia causes downregulation of mismatch repair system in cells[33–35] (Fig. 3c). In addition, we also revealed that significant PAS up-regulation of factor 6 associated-pathway "Glycolysis" in MMRd patients, similar to the results obtained by Pelka et al.[22] (Fig. 3c). Overall, the pathway-centric interpretation of MMR status-related multicellular modules substantiated their validity and trustworthiness.

To test the robustness and generality of the multicellular pathway modules uncovered by scPAFA, we used the CRC bulk RNA dataset to construct an MMR status classifier based on the high-weight pathways associated with the modules. We applied a support vector classification (SVC) model with L1 regularization, using gene set variation analysis (GSVA) and single-sample gene set enrichment analysis(ssGSEA) score of high-weight pathways as features (Fig. 3d). Based on GSVA score, the L1 regularization parameter was set as 0.2 based on the area under the receiver operating characteristic curve (AUROC) from 4-fold cross-validation in the training set (Fig. 3e). The AUROC of MMR status classifier achieved 0.91 in the test set (GSE39582[36]) and independent test set (TCGA COAD_READ) (Fig. 3f), besides, the features remained after L1 regularization included pathways such as "METAPROGRAM_EPITHELIAL_INTERFERON_MHC", "METAPROGRAM_6_HYPOXIA" and "Glycolysis" (Fig. 3g). Additionally, classifiers based on ssGSEA scores also lead to consistent results (Supplementary Fig. 1c–f). The above results showed the remarkable ability of scPAFA in feature selection.

## scPAFA characterized intra-patient heterogeneity in CRC

MMR status represents a form of inter-patient heterogeneity. Furthermore, we utilized scPAFA to characterize the differences between CRC tumors and adjacent colon tissues, as well as between primary and metastatic sites, thereby highlighting intra-patient heterogeneity. We collected scRNA-seq data of primary colorectal tumor (primary tumor), adjacent colon (primary normal), liver metastases (metastasis tumor), and adjacent liver (metastasis normal) from Liu et al.[23] and merged into a CRC-CRC liver metastasis(CRC-CRLM) dataset (Methods). This CRC-CRLM dataset consists of

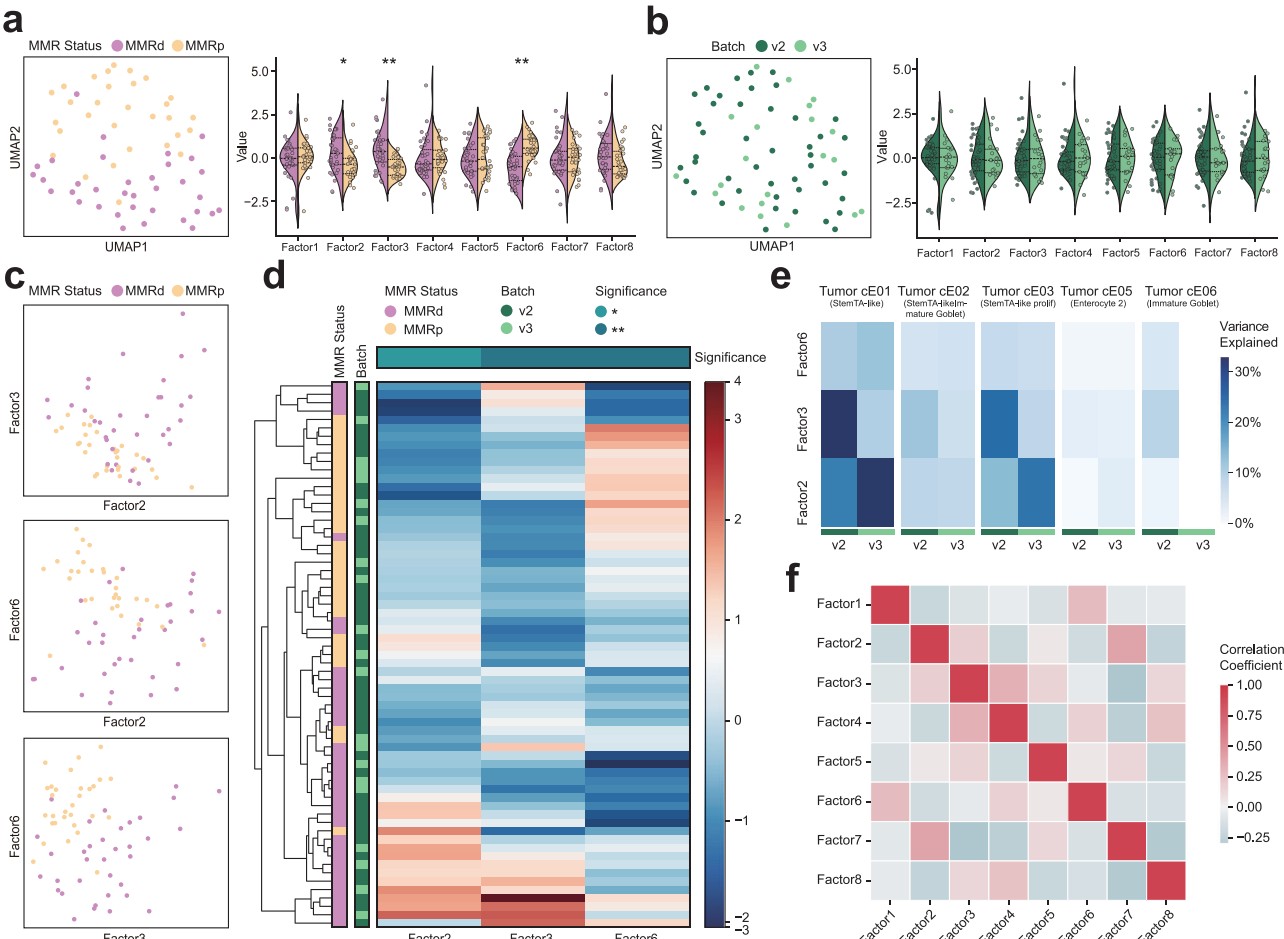

**Fig. 2 | Multicellular pathway modules stratify CRC patients. a** UMAP of latent factor matrix shows stratification between MMRp and MMRd samples. Violin plots show the difference in factor values between MMRp and MMRd samples using the Mann-Whitney U test ($n = 65$ pseudobulk samples). The dashed lines represent the median (center line), first quartile (lower line), and third quartile (upper line) of the data distribution. * adjust $p$-value < 0.05, **adjust $p$-value < 0.01. **b** UMAP of latent factor matrix shows uniform distribution of samples from different batches. Violin plots show the difference in factor values between batch v2 and batch v3 samples using the Mann-Whitney U test ($n = 65$ pseudobulk samples). The dashed lines represent the median (center line), first quartile (lower line), and third quartile

(upper line) of the data distribution. * adjust $p$-value < 0.05, **adjust $p$-value < 0.01. **c** Scatter plots of factors 2, 3, and 6 show stratification between MMRp and MMRd samples. **d** Heatmap with hierarchical clustering of factors 2, 3, and 6 shows stratification between MMRp and MMRd samples. The annotation of columns shows the difference in factor values between MMRp and MMRd samples using the Mann-Whitney U test ($n = 65$ pseudobulk samples). * adjust $p$-value < 0.05, **adjust $p$-value < 0.01. **e** The heatmap displays the percentage of variance explained by each factor across the different groups (batch) and views (cell type). For simplicity, only factors 2, 3, and 6 were shown. **f** The heatmap displays the correlation coefficient between different factors.

40 samples from 10 patients, including 163,347 cells which were annotated as 80 cell clusters. After applying the scPAFA framework (Methods), the result provided a latent factor matrix with 8 factors that explain at least 1% of the variance. The Kruskal-Wallis test revealed significant differences in factors 4, 5, and 6 across the four sample sites (Fig. 4a). Consistently, UMAP visualization and hierarchical clustering heatmap demonstrated clear stratification of samples from primary tumor/normal and metastatic tumor/normal groups (Fig. 4b, c). The Mann-Whitney U Test further revealed significant differences in factors 4 and 6 between the primary and metastatic sites (Fig. 4d), while factor 5 showed significant differences between tumor and normal tissues (Fig. 4e). These findings suggest that factors 4 and 6 represent metastasis-related multicellular pathway modules, while factor 5 represents malignant-related multicellular pathway modules. Liu et al.[23] reported that SPP1[+] macrophages (hc54_Mph−SPP1) were predominant in liver metastasis, indicating its pro-metastasis role, while DC3s (hC38_cDC2−C1QC) were characterized as malignancy driven[23]. Consistently, in the scPAFA results, the percentage of variance explained by the metastasis-related factor 6 is highest in hc54_Mph−SPP1, whereas the percentage of variance explained by the malignancy-related factor 5 in hC38_cDC2−C1QC is the second highest among all cell types (Fig. 4f).

Furthermore, factor 4 accounts for the highest percentage of variance explained in hC72_B-ILGC3, while factor 5 explains the highest percentage of variance in hC74_B-MK167, the result suggested the importance of hC72_B-ILGC3 and hC74_B-MK167 in characterizing CRC-CRLM heterogeneity, which was not mentioned by Liu et al.[23]. The feature weights matrixes corresponding to factors 4, 5, and 6 can be used to interpret these multicellular pathway modules (Supplementary Fig. 2a–c). For example, the value of factor 5 was significantly lower in tumors than normal (Fig. 4e), thus negative high-weight pathway-cell type pairs of factor 5 can be considered as malignant characteristics. In factor 5, the top 3 negatively weighted pathway-cell type pairs were all related to the transcription factor nuclear factor kB (NF-kB) signal (Supplementary Fig. 2b), which has been recognized as a critical factor in the initiation and propagation of CRC[37,38].

## Identification of SLE-related multicellular pathway modules

In addition to cancer, we applied scPAFA to the SLE dataset. SLE is a heterogeneous autoimmune disease, previous bulk transcriptomic profiling reported that increased type 1 interferon signaling, dysregulated lymphocyte activation, and failure of apoptotic clearance as hallmarks of SLE[3]. For the large-scale lupus scRNA-seq atlas, we extracted 1,067,499 cells from 3

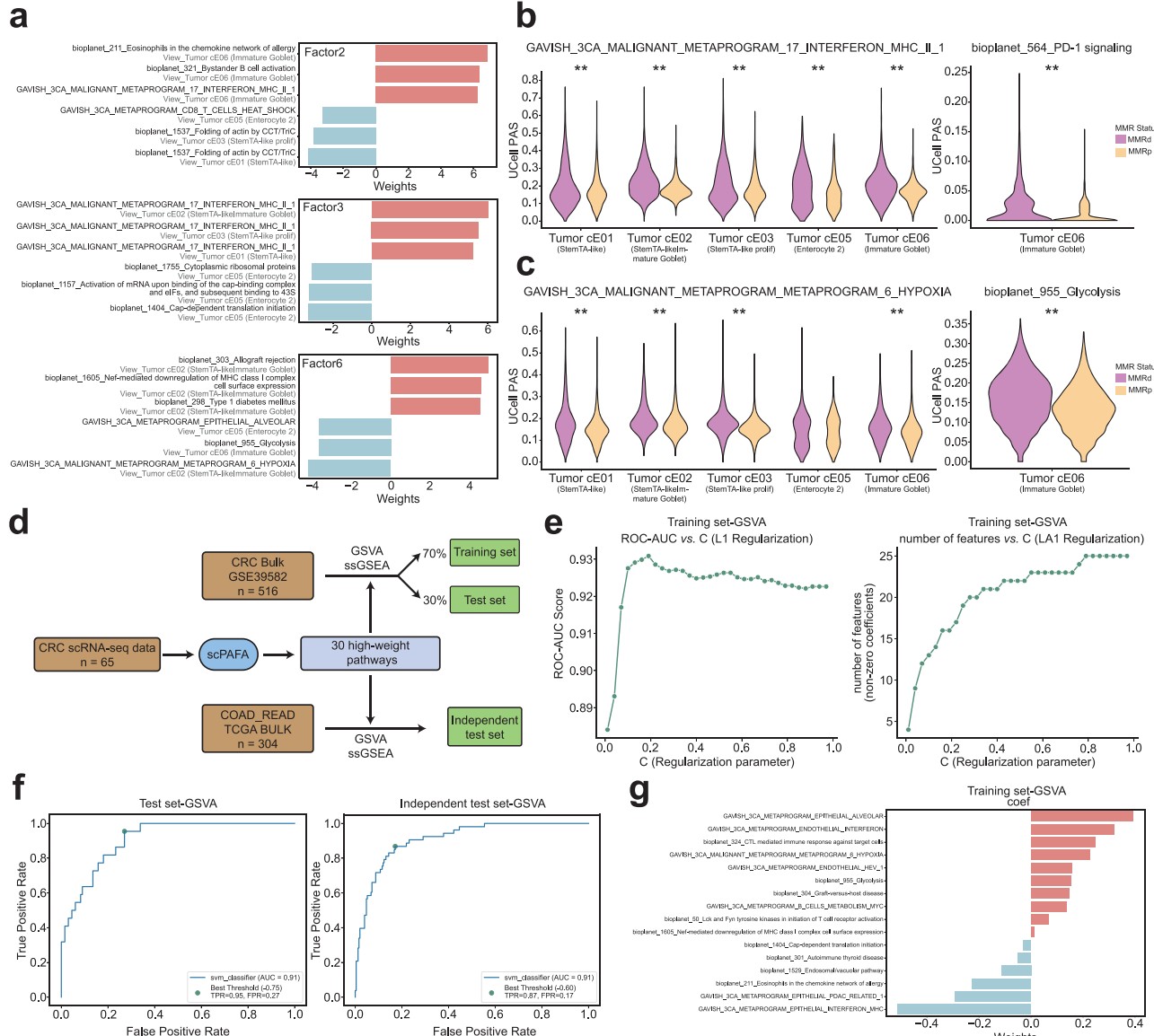

**Fig. 3 | Pathway-centric interpretation and classifier training based on MMR status-related modules. a** Butterfly bar plot displaying the pathway-cell type pairs with the top 3 positive and negative weights of factors 2, 3, and 6. Violin plots show the difference of cell-level PAS of "METAPROGRAM_17_INTERFER-ON_MHC_II_1" (**b**), "PD-1 signaling" (**b**), "METAPROGRAM_6_HYPOXIA" (**c**) and "Glycolysis" (**c**) between MMRp and MMRd using Mann-Whitney U test ($n = 49{,}330$; 9186; 38,012; 3043 and 4218 cells for the Tumor cE01, Tumor cE02, Tumor cE03, Tumor cE05 and Tumor cE06, respectively). *$p$-value < 0.05, **$p$-value < 0.01. **d** Flowchart for classifier training on bulk RNA data. **e** Line charts illustrate the impact of various regularization parameter values on AUROC and the number of features in the training set (4-fold cross-validation). Based on GSVA score. **f** The AUROC of the classifier on the test set and an independent test set. Based on GSVA score. **g** Butterfly bar plots display the coefficients of features included in the classifier. Based on GSVA score.

processing batches that simultaneously contain SLE and health controls to Identify SLE-related multicellular pathway modules by scPAFA workflow. Single-group MOFA framework discovered batch effects in samples' stratification caused by processing batches (Supplementary Fig. 3a). It's noteworthy that a cluster (cluster 0) of pseudobulk samples from batch 3 separated from other pseudobulk samples, which was characterized by multicellular pathway modules associated with red blood cells (Supplementary Fig. 3b–d). Considering that the lupus dataset was collected from PBMCs, we speculate that cells originating from this cluster have been contaminated; therefore, we excluded these cells, leaving 941,542 cells remaining. The remained dataset was then split into a training set and a test set with a 7:3 ratio based on pseudobulk samples. For the training set, the PAS matrix was reformatted with 3 processing batches, 175 pseudobulk samples, and 7 cell types as input for the multi-group MOFA+ framework. The UMAP plot of the latent factor matrix with 8 factors that explain at least

1% of variance showed distinct stratification between SLE patients and healthy control, while pseudobulk samples from different batches exhibit a uniform distribution (Fig. 5a, b). Moreover, the value of factor 1 was significantly increased in SLE patients, while factor 6 was significantly decreased (Fig. 5a), hence factors 1 and 6 were identified as SLE-related multicellular pathway modules. Consistently, obvious stratification between SLE patients and healthy control can also be observed in two-dimensional scatter plots and hierarchical clustering heatmap based on the values of factors 1 and 6 (Fig. 5c, d). Compared to other cell types, factor 1 explained a higher proportion of variance in classical monocytes (cM) and non-classical monocytes (ncM), while factor 6 explained a higher proportion of variance in CD4 + T cells (T4) and CD8 + T cells (T8) (Fig. 5e). Besides, factor 1 negatively correlates with factor 6 (Fig. 5f). SLE-related multicellular pathway modules consist of "Interferon alpha/beta signaling" (type 1 interferon) in all cell types, "Type II interferon signaling (interferon-gamma)" in ncM

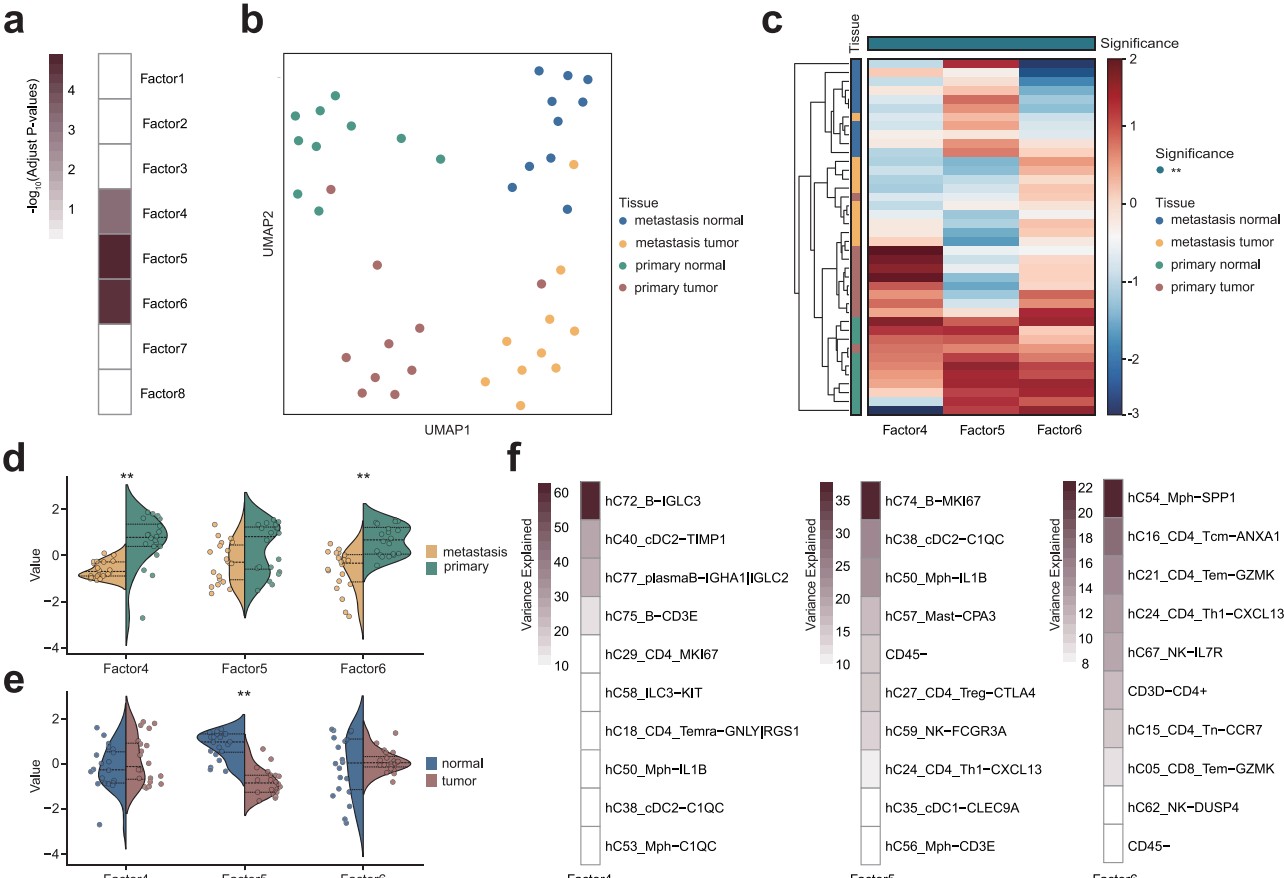

**Fig. 4 | scPAFA captures heterogeneity in CRC-CRLM dataset. a** The heatmap displays the -log10(adjusted p-value) from the Kruskal-Wallis test (*n* = 40 pseudo-bulk samples), illustrating the differences in factor values across samples from primary colorectal tumors (primary tumor), adjacent colon (primary normal), liver metastases (metastasis tumor), and adjacent liver (metastasis normal). **b** UMAP based on the value of factors 4, 5, and 6 shows stratification among 4 sample sites. **c** Heatmap with hierarchical clustering of factors 4, 5, and 6 shows stratification among 4 sample sites. The annotation of columns shows the difference in factor values among 4 sample sites. using the Kruskal-Wallis test (*n* = 40 pseudo-bulk samples). * adjust *p*-value < 0.05, **adjust *p*-value < 0.01. **d** Violin plots show the difference in factor values of factors 4, 5, and 6 between samples from the primary site and metastasis site using the Mann-Whitney U test (*n* = 40 pseudobulk samples). The dashed lines represent the median (center line), first quartile (lower line), and third quartile (upper line) of the data distribution. * adjust p-value < 0.05, **adjust *p*-value < 0.01. **e** Violin plots show the difference in factor values of factors 4, 5, and 6 between samples from tumor and adjacent normal using the Mann-Whitney U test (*n* = 40 pseudobulk samples). The dashed lines represent the median (center line), first quartile (lower line), and third quartile (upper line) of the data distribution. * adjust *p*-value < 0.05, **adjust *p*-value < 0.01. **f** The heatmap displays the percentage of variance explained by factors 4, 5, and 6 across different cell types, highlighting the top 10 cell types for each factor.

and classical dendritic cell (cDC), "Cross-presentation of particulate exogenous antigens (phagosomes)" in cM and "Granzyme A-mediated apoptosis pathway" in T8, etc. (Fig. 6a, Supplementary Data 3). Conformingly, PAS of "Interferon alpha/beta signaling" and "Type II interferon signaling (interferon-gamma)" was significantly elevated across all cell types, especially in cDC and monocytes (cM,nCM) (Fig. 6b, c). Moreover, PAS of "Cross-presentation of particulate exogenous antigens (phagosomes)" was significantly increased in cM (Fig. 6d), which could promote the activation of cytotoxic CD8 + T cells[39], correspondingly, PAS of "Granzyme A-mediated apoptosis pathway" was significantly upregulated in T8 of SLE patients (Fig. 6e).

To further validate the reliability of the modules, we train the SLE/healthy SVC classifier using SLE-related multicellular pathway modules separately on scRNA-seq data and bulk RNA data. For scRNA-seq data, we used pseudobulk-PAS of 78 pathway-cell type pairs as features, which were the top-40 pathway-cell type pairs with the highest absolute weights associated with factors 1 and 6, respectively (Fig. 6f). After 4-fold cross-validation in training set, we set L1 regularization parameter as 0.05 for feature reduction (Fig. 6g), the AUROC of classifier achieved 0.91 in test set, which contained 10 pathway-cell type pairs (Fig. 6h, i). For bulk RNA data, 78 pathway-cell type pairs were aggregated into 52 pathways as features, 70% of

samples from GSE88884[40] were assigned as bulk training set, while 30% of samples from GSE88884 and all samples from GSE61635 were assigned as bulk test set 1 and 2 (Fig. 6f). Following training like abovementioned process, the AUROC of the 5-features-classifier based on GSVA score including 'Interferon alpha/beta signaling' achieved 0.93 and 0.98 in bulk test set 1 and 2, separately (Fig. 6j–l); consistent results were also found in classifier based on ssGSEA score (Supplementary Fig. 3e–h).

**Comparison of SCPA and scPAFA**

SCPA is a specialized tool for pathway analysis in scRNA-seq data that evaluates the multivariate distributions of pathways to determine their significance difference across conditions[11]. Instead of calculating PAS, it provides a Qval that reflects the variation in pathway activity. SCPA can find the variation of pathway activity across conditions within each cell type, and provide a cell type-pathway Qval matrix. We compared the computational efficiency and the ability to detect pathway activity differences between SCPA and scPAFA on CRC and lupus datasets ("Methods"). SCPA uses a down-sampling strategy (selecting 500 cells per condition per cell type as default) in its main function 'compare_pathways' to reduce the computational load, we run SCPA by downsampling to 500, 1000, and 2000 cells, respectively. We observed that as the number of downsampled cells increased, the computation

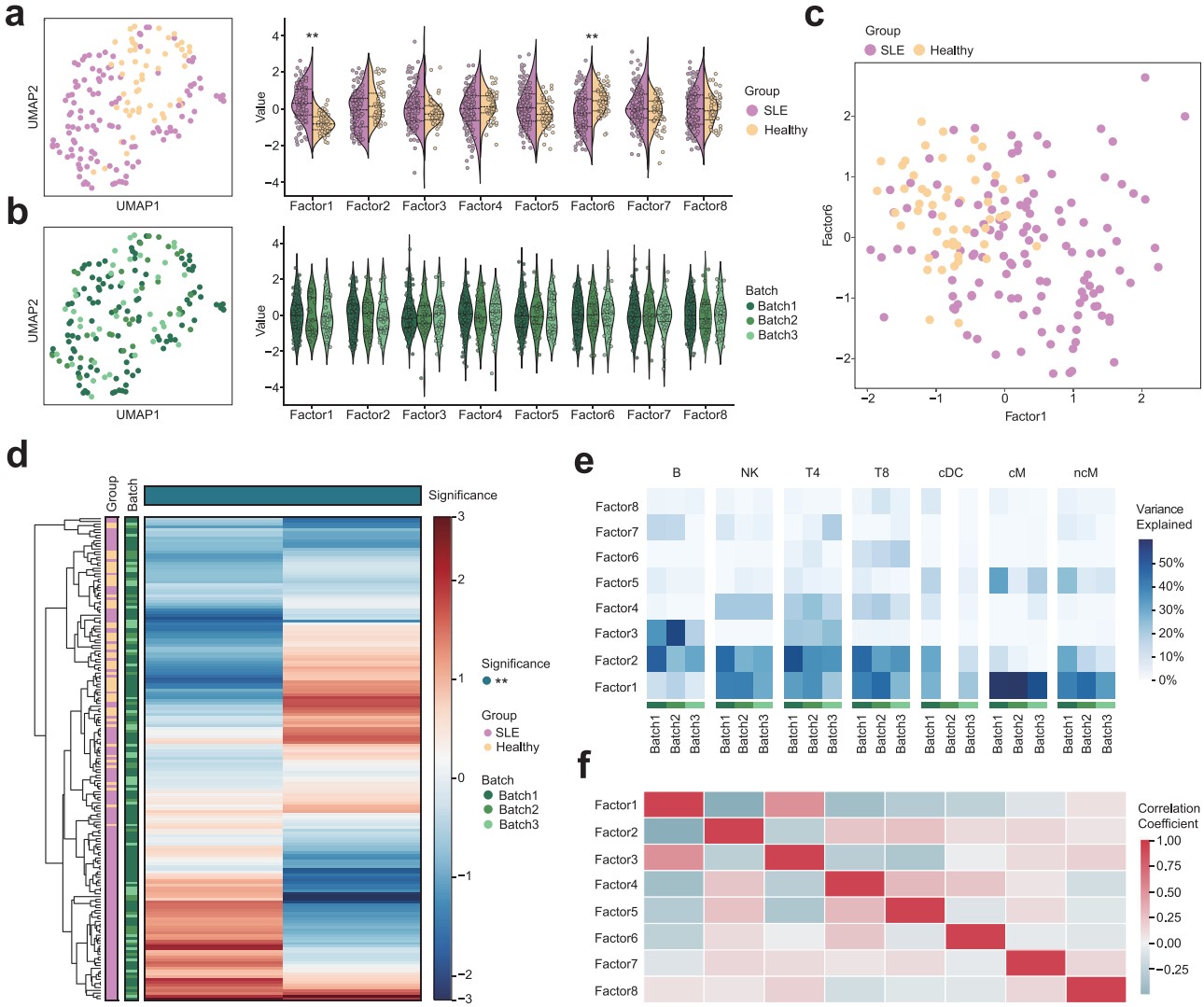

**Fig. 5 | Multicellular pathway modules distinguish SLE patients from healthy controls. a** UMAP of latent factor matrix shows stratification between SLE patients and healthy controls. Violin plots show the difference in factor values between SLE patients and healthy controls using the Mann-Whitney U test ($n = 175$ pseudobulk samples). * adjust $p$-value < 0.05, **adjust $p$-value < 0.01. **b** UMAP of latent factor matrix shows uniform distribution of samples from different batches. Violin plots show the difference in factor values between batches using the Mann-Whitney U test ($n = 175$ pseudobulk samples). * adjust p-value < 0.05, **adjust $p$-value < 0.01.

**c** Scatter plots of factors 1 and 6 show stratification between SLE patients and healthy controls. **d** Heatmap with hierarchical clustering of factors 1 and 6 shows stratification between SLE patients and healthy controls. The annotation of columns shows the difference in factor values between SLE patients and healthy controls using the Mann-Whitney U test ($n = 175$ pseudobulk samples). * adjust $p$-value < 0.05, **adjust $p$-value < 0.01. **e** The heatmap displays the percentage of variance explained by each factor across the different groups (batch) and views (cell type). **f** The heatmap displays the correlation coefficient between different factors.

time for SCPA grew dramatically (Fig. 7a, b). In contrast, scPAFA was approximately 1.3(500 cells in the lupus dataset) to 103.9 (2000 cells in the CRC dataset) times faster than SCPA, even when scPAFA calculated PAS for all cells (Fig. 7a, b and Supplementary Data 1). To produce results comparable to those from SCPA, we combined the weight matrices of disease-related factors from scPAFA to generate a cell type-pathway weight matrix (Methods). The cell type-pathway Qval matrix (SCPA) and the cell type-pathway weight matrix (scPAFA) were then filtered to retain only the union of the top 3 pathways in each cell type, and subsequently sorted in descending order based on the mean Qval/weight of all cell types.

In the CRC dataset, scPAFA emphasizes the most notable differential pathway between MMRp and MMRd is "METAPROGRAM_17_INTERFERON_MHC_II_1"(Fig. 7c), which was consistent with the gene program reported in the original study of Pelka et al.[22]. However, this pathway was not emphasized in the SCPA result (Fig. 7c and Supplementary Fig. 4a). It was widely reported that increased type 1 interferon (interferon-alpha/beta) signaling is the hallmark of SLE[3,40]. Consistently, in lupus

dataset, scPAFA emphasizes the most notable differential pathway is "Interferon alpha/beta signaling", which was also detected by SCPA (2000 cells) (Fig. 7d). However, this pathway was not emphasized in the result of SCPA (500 cells) (Supplementary Fig. 4b), indicating downsampling strategy of SCPA could result in the loss of information in large datasets. In brief, compared with SCPA, scPAFA can more efficiently utilize data from all cells, avoiding potential information loss and thereby detecting reliable differential pathways in large-scale datasets.

## Discussion

In omics research, pathways are widely considered as informatics and functional biological units based on prior knowledge. Pathway analysis methods have been extensively employed to help researchers identify key biological themes for comprehending transcriptome[41,42]. scPAFA can rapidly compute PAS for a large number of pathways on large-scale scRNA-seq data, allowing for efficient re-representation of transcriptome data based on prior knowledge. The "fast_ucell" function can provide the same PAS of UCell,

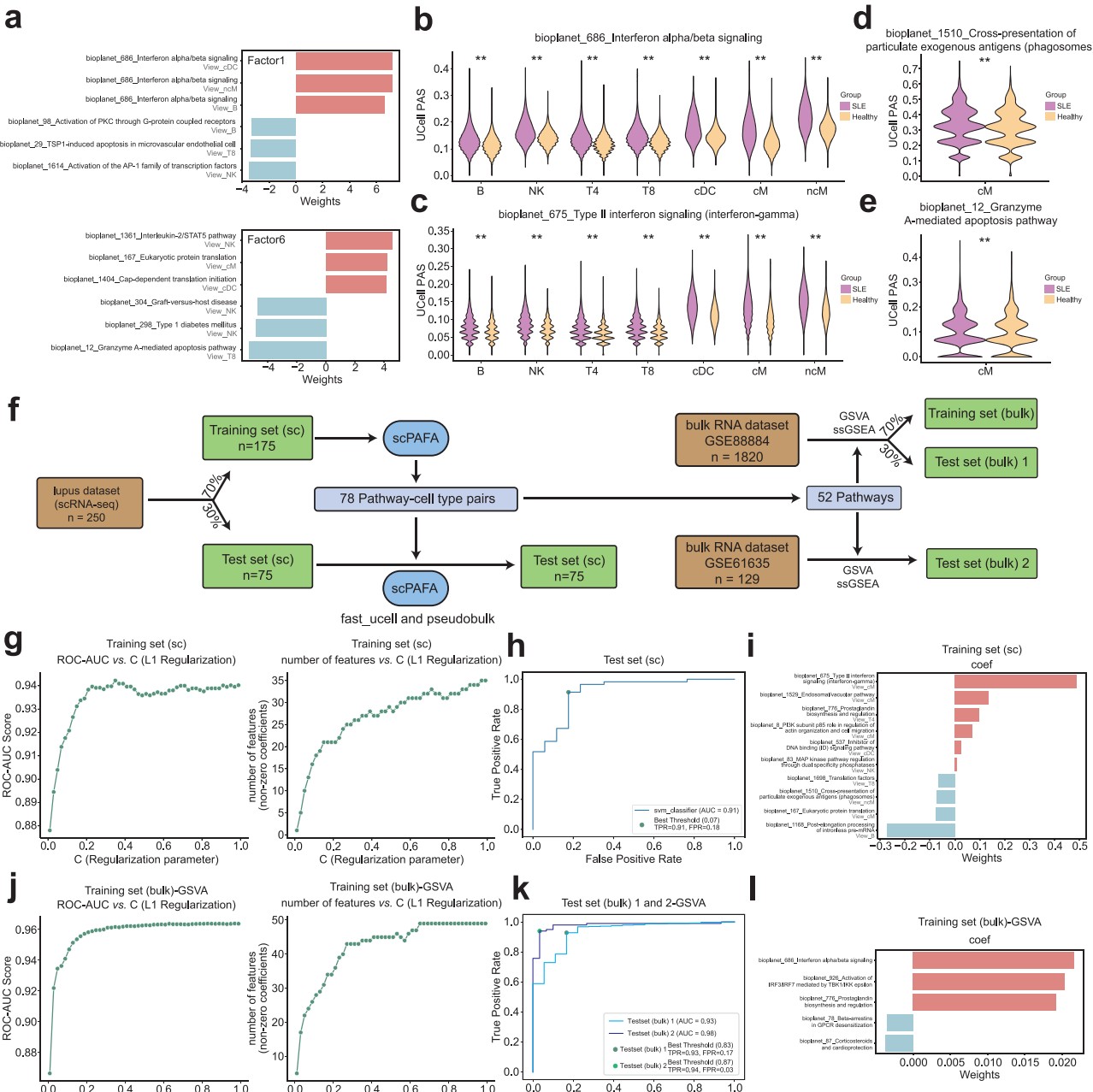

**Fig. 6 | Pathway-centric interpretation and classifier training based on SLE-related modules. a** Butterfly bar plots displaying the pathway-cell type pairs with the top 3 positive and negative weights of factors 1 and 6. Violin plots show the difference in cell-level PAS of 'Interferon alpha/beta signaling' (**b**), 'Type II interferon signaling (interferon-gamma)' (**c**), 'Cross-presentation of particulate exogenous antigens (phagosomes)' (**d**) and Granzyme A-mediated apoptosis pathway' (**e**) between SLE patients and healthy controls using the Mann-Whitney U test ($n$ = 189,897; 168,032; 131,317; 82,238; 47,231; 27,813; and 9685 cells for the T4, cM, T8, B, NK, ncM, and cDC cell types, respectively). *$p$-value < 0.05, **$p$-value < 0.01. **f** Flowchart for classifier training on scRNA-seq data and bulk RNA data. **g** Line

charts illustrate the impact of various regularization parameter values on AUROC and the number of features in the single-cell training set (4-fold cross-validation). **h** The AUROC of the classifier on the single-cell test set. **i** Butterfly bar plots display the coefficients of features included in the classifier trained on single-cell data. **j** Line charts illustrate the impact of various regularization parameter values on AUROC and the number of features in the bulk training set (4-fold cross-validation). Based on GSVA score. **k** The AUROC of the classifier on the bulk test set and bulk independent test set. Based on GSVA score. **l** Butterfly bar plots display the coefficients of features included in the classifier trained on bulk data. Based on GSVA score.

which is closely related to AUCell PAS by setting the same max rank threshold. Similar to AUCell, the PAS of "fast_ucell" is based on the ranking of gene expression within each cell, making them insensitive to the dataset in which the cells are located. Considering the "AddModuleScore" function provided by Seurat ("score_genes" in Scanpy) is widely used, we also provide a parallelized implementation "fast_score_genes" in scPAFA. Unlike "fast_ucell", the PAS of "fast_score_genes" is sensitive to dataset composition. For PAS generating, we primarily used NCATS BioPlanet [27] with scPAFA,

an integrated resource of 1658 manually curated human pathways. Users can also choose other pathway databases or customize pathway sets for the specific biological context of the scRNA-seq dataset. For instance, we also employed 3CA metaprogram [28] on the CRC datasets for interpretation based on prior knowledge in oncology.

Cell type annotation is a key step in the analysis of scRNA-seq data, thus integrating transcriptomic features across different cell types to facilitate interpretation at the level of biological conditions is an issue worthy of

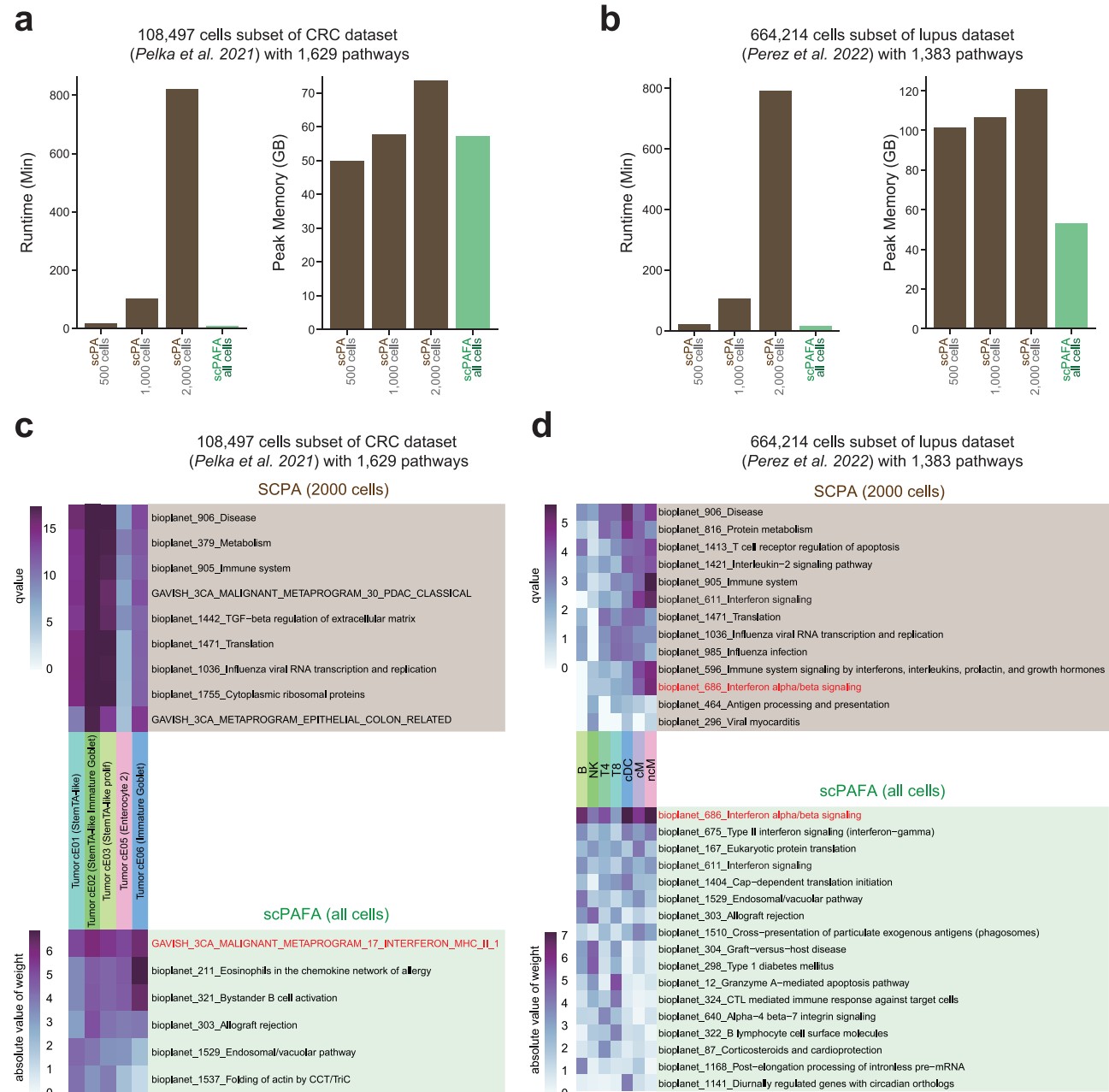

**Fig. 7 | Comparison of SCPA and scPAFA.** Bar plots show the runtime and the peak memory usage of scPA (10 cores) and scPAFA (10 cores) on CRC (**a**) scRNA-seq dataset (108,497 cells subset with 1629 pathways) and lupus (**b**) scRNA-seq dataset (664,214 cells subset with 1383 pathways). The heatmaps highlight the most prominent (the union of top 3 pathways in each cell type) MMR status-related (**c**) and SLE-related (**d**) differential pathways identified by both the SCPA (2000 cells) and scPAFA (all cell) methods. The pathway highlighted in red is the one expected to be identified.

attention. Several computational methodologies focused on multicellular integration have emerged: DIALOGUE[43] was developed to systematically uncover combinations of coordinated cellular programs in different cell types from either spatial data or scRNA-seq data; Tensor-cell2cell[44] can reveal context-dependent communication patterns linked to various phenotypic states, influenced by distinct combinations of cell types and ligand-receptor pairs; MOFAcellulaR[16] allows the integration of measurements of independent single-cell, spatial, and bulk datasets to contextualize multicellular responses in disease. Inspired by the methods mentioned above, especially MOFAcellulaR[16], we reutilized the MOFA framework in scPAFA for multicellular integration. scPAFA can be regarded as a complement to existing pathway analysis methods in scRNA-seq data, aiming to identify interpretable pathway-based multicellular transcriptomic features associated with the biological conditions of interest.

In this study, scPAFA was applied to CRC and lupus datasets, identifying multicellular pathway modules that capture the heterogeneity of CRC and transcriptional abnormalities in SLE. These modules are consistent with previous reports in biological interpretation and can further be dissected to extract features for training classifiers. We compared scPAFA with the recently proposed method SCPA. SCPA identifies differential pathways under various conditions within each cell type, a task that scPAFA can also accomplish. However, scPAFA is faster than SCPA and avoids the information loss associated with downsampling. Additionally, scPAFA can provide modules that integrate differential information across multiple cell types. However, scPAFA also has several limitations. Most importantly, scPAFA requires studies to include at least 15 samples (not cells), which is the requirement of using the MOFA model for identifying multicellular pathway modules. In studies with samples less than 15, SCPA is more

flexible and applicable than scPAFA. It is worth mentioning that scPAFA can also be used solely for the rapid calculation of PAS, in which case the sample size does not impact the results. The other limitations of scPAFA include that the selection of pathways is a supervised process, where using a redundant pathway set may increase the complexity of the results, while choosing too few or pathways that do not align with the biological context may cause information loss. Moreover, pseudobulk integration of the PAS matrix would lead to information loss since the PAS of multiple cells is aggregated. Besides, MOFA is a linear model, interpretability is achieved by sacrificing information content in each factor, and the interpretability may also decrease with multiple disease states. In addition, the pathways and cell types within the same module can only suggest data-driven associations, rather than determining biological causal effects.

In summary, we present scPAFA, an efficient and user-friendly tool for researchers to generate PAS and uncover biologically interpretable disease-related multicellular pathway modules from scRNA-seq data. scPAFA is compatible with typical single-cell analysis workflows, which can integrate variance captured from different cell types and enhance understanding of disease biology.

## Methods

### Input of scPAFA

The input for scPAFA includes AnnData class (built jointly with Scanpy) containing single-cell gene expression matrix and cell-level metadata, a Python dictionary containing pathways names as keys and gene symbols in pathways as values, and a pandas dataframe containing sample-level metadata (clinical information). We provided a function "generate_pathway_input" to filter and reformat pathway sets dictionary for subsequent analysis based on the number of overlapping genes between pathways and the gene expression matrix.

### Computational optimizations of "fast_ucell"

The function "fast_ucell" is a Python implementation of the R package UCell[9], which is faster, with similar adjustable parameters, capable of yielding consistent PAS. UCell calculates gene signature scores for scRNA-seq data based on Mann-Whitney U statistic, which is closely related to methods based on AUC scores such as AUCell[10]. UCell mainly consists of two steps: 1) Given single-cell gene expression matrix $X$ with genes as $g$ and cells as $c$ (e.g., count matrix or log-normalized count matrix), relative gene expression ranks matrix $R$ is calculated by sorting all genes in descending order in each cell. To mitigate the uninformative tail caused by the sparsity of single-cell data, $r_{c,g} = r_{max} + 1$ for all $r_{c,g} > r_{max}$, with $r_{max} = 1500$ by default. 2) For a pathway $S$ composed of $n$ genes ($S_1, \ldots, S_n$), PAS of each cell $j$ in $X$ was calculated with the formula, ranging from 0 to 1:

$$PAS_j = 1 - \frac{U_j}{n * r_{max}}$$

where $U_j$ is the Mann-Whitney U statistic calculated by:

$$U_j = \sum_{i=1}^{n} r'_{j,i} - \frac{n(n+1)}{2}$$

and $R'_j$ is obtained by sub-setting $R_j$ on the pathway $S$.

The performance of "fast_ucell" running on large datasets was enhanced by multi-process parallelism and vectorized computation based on the original UCell computation process. In step 1, single-cell gene expression matrix $X$ in sparse matrix format from large dataset is automatically split into chunks of reduced size (default as 100,000 cells), the relative gene expression ranks matrix $R$ is calculated parallelly by using "stats.rankdata" function in SciPy (v1.11.3) package, then $R$ is reformatted into sparse matrix by assigning $r_{c,g} = 0$ for all $r_{c,g} > r_{max}$. In step 2, $R_{sub}$ is extracted from $R$, which consists of the union of genes contained in all pathways in pathway set $P$, then $R_{sub}$ is split into chunks by cells (default as 100,000 cells) for serial processing while pathway set $P$ is split into chunks by

pathways for parallel processing. For a pathway $S$ composed of $n$ genes, the PAS of $S$ in a chunk of $R_{sub}$ is calculated by a vectorized computation process. In brief, $R_S$ is extracted from the chunk of $R_{sub}$, which only includes all genes in $S$; row (cell)-wise summation of the $R_S$ is performed and stored as vector $V$, and cells with a sum of 0 are labeled as $C_0$, while others are labeled as $C_1$. For cells in $C_0$, the PAS of $S$ is approximately specified as 0. For cells in $C_1$, the numbers of zero columns(gene) in $R_S$ are counted as vector $z$, then PAS of $S$ is calculated with the formula using SciPy package:

$$PAS_{C_1} = 1 - \frac{V_{C_1} + (r_{max} + 1) * z - n(n+1)/2}{n * r_{max}}$$

To handle a large number of pathways, the pathway set $P$ is divided and distributed across multiple cores for parallel processing using "ProcessPool" from Pathos (v0.3.1).

### Computational optimizations of "fast_score_genes"

The function "fast_score_genes" is a multiprocessing implementation of the "score_genes" function in Scanpy ("AddModuleScore" function in Seurat). In function "score_genes", PAS of a gene signature $S$ in a given cell $j$ of single-cell gene expression matrix $X$ is defined as the average relative expression of the overlap genes of $j$ in $S$, which can be calculated by a three-step process: 1) using the mean expression across all input cells to categorize all input genes into bins (default set to 25 bins) of uniform size, 2) randomly choosing reference genes (default as 50 genes) from the same expression bin for each gene in the gene signature $S$ as reference gene set $S_{ref}$. 3) the PAS of cell $j$ is the average expression of $S$ subtracted with the average expression of reference gene set $S_{ref}$:

$$PAS_j = \sum_{g \in S} X_{j,g} / n - \sum_{g_{ref} \in S_{ref}} X_{j,g_{ref}} / n_{ref}$$

where $X_{j,g}$ is the expression of gene $g$ in cell $j$, $n$ and $n_{ref}$ is the number of genes in $S$ and $S_{ref}$, respectively. The computations in Step 3 are supported by vectorization processes in the SciPy and NumPy packages. In "fast_score_genes", given a single-cell gene expression matrix $X$, step 1 was performed first, then $X$ can be split into smaller chunks by cells (default as 100,000 cells) for serial processing (minimizing memory usage). For each chunk of $X$, steps 2 and 3 can be executed concurrently using multiple pathways in pathway set $P$. Similar to "fast_ucell", the pathway set $P$ is divided and distributed across multiple cores for parallel processing using "ProcessPool" from Pathos.

### Details of pseudobulk integration functions

We developed "generate_scpafa_input" and "generate_scpafa_input_multigroup" functions for pseudobulk integration with PAS matrix and cell-level metadata. Users can specify the column names in cell-level metadata corresponding to sample information, cell type, and batch information as well as quality control metrics such as the cell number threshold of a pseudobulk sample, the sample number threshold, and the sample proportion threshold within groups of a cell type. These functions generate a long-format table, which is the input for MOFA containing 5 columns: column "sample" including sample information, column "group" including batch information, column "feature" including names of pathways, column 'view' including cell type information and column "value" including pseudobulk PAS values.

### Wrapper function for MOFA model training

We integrated the source code from mofapy2 (v0.7.0) with modifications to support the new version of the pandas (≥v2.0) package. For single-click runnable MOFA model training, we provided wrapper function "run_mofapy2" based on the "entry_point" class of mofapy2.

### Extracting the latent factor matrix and weight matrix from the converged MOFA model. Python package mofax (v0.3.6) is employed for extracting the latent factor matrix and weight matrix from the converged

MOFA model. The MOFA model is imported using "mofa_model" function. The "get_factors" function with the setting "df=True, concatenate_groups =False, scale=True" extracts the latent factor matrices and scaled per factor per group, then these matrices are integrated into a single latent factor matrix using "concat" function of Pandas (v2.1.1) package.

Similarly, the "get_weights" function with the setting "df=True, concatenate_views =False, scale=True" extracts the weight matrices and scaled per factor per cell type (view), then these matrices are integrated into a single weight matrix using "concat" function of Pandas package.

### Identification of disease-related multicellular pathway modules (latent factors)

We assumed that multicellular pathway modules (latent factors) with values differing across different disease conditions are related to the disease, to test the significance of differences in factor values between disease states, we provided the "parametric_test_category" and the "nonparametric_test_category" functions in scPAFA. These functions automatically detect the number of disease state conditions. If there are two conditions, a Mann-Whitney U test (non-parametric) or t-test (parametric) is used; if there are more than two conditions, a Kruskal-Wallis test (non-parametric) or ANOVA (parametric) is applied. The p-value from these tests was then adjusted by using Benjamini-Hochberg correction across all factors. Factors with adjusted p-values less than 0.05 can be considered related to the disease. Besides, we also provided the "cal_correlation" function to identify modules associated with continuous traits.

### Visualization functions

For visualization, the "runumap_and_plot", "plot_factor_scatter_2D", and "draw_cluster_heatmap" functions are used to show the stratification of samples, while the "plot_weights_butterfly" function can be used to show features with top absolute weights.

### Single-cell RNA-seq data preprocessing

For CRC dataset[22], the single-cell gene expression matrix and metadata were downloaded from Broad Institute's Single Cell Portal (https://singlecell.broadinstitute.org/single_cell/study/SCP1162). We used Scanpy (v1.9.5) to build AnnData, only genes with gene ID starting with "ENSG" were retained, then after filtering out genes expressed in fewer than 30 cells, the dataset comprises 371,223 cells and 26320 genes.

For CRC-CRLM dataset[23], the single-cell gene expression matrix and metadata of primary colorectal tumor, adjacent colon, liver metastases and adjacent liver were downloaded from Gene Expression Omnibus (GEO) with an accession number of GSE164522, the dataset comprises 163,347 cells and 24662 genes. For lupus dataset[3], AnnData in h5ad format was downloaded from CZ CELLxGENE Annotate(https://cellxgene.cziscience.com/collections/436154da-bcf1-4130-9c8b-120ff9a888f2), after filtering out genes expressed in fewer than 30 cells, the dataset includes 1,263,676 cells and 20514 genes.

### Bulk RNA data preprocessing

Microarray gene expression matrix and corresponding sample-level metadata of GSE39582, GSE61635, and GSE88884 were collected using GEOparse(v2.0.3). Bulk RNA-seq gene expression matrix and corresponding sample-level metadata of TCGA Colon and Rectal Cancer (COAD-READ) dataset were collected from UCSC Xena(https://xena.ucsc.edu/), and samples from primary solid tumor were used for further analysis. After filtering out samples with missing clinical information, the TCGA COAD-READ dataset comprises 251 MMRp samples and 53 MMRd samples; GSE39582 includes 444 MMRp samples and 72 MMRd samples; GSE61635 contains 30 healthy controls and 99 SLE samples, while GSE88884 includes 60 healthy controls and 1760 SLE samples.

### Runtime and memory usage evaluation

We used an Intel X79 server (Ubuntu 20.04) with E5-2670v2 CPU and 256GB RAM for runtime and memory usage evaluation. For the CRC dataset, we sampled 100 thousand, 200 thousand, and all cells from the dataset as input; the pathways from NCATS BioPlanet collection and 3CA metaprogram with more than 6 overlap genes (1629 pathways) were used as pathway input. For the lupus dataset, we sampled 100 thousand, 200 thousand, 500 thousand, and all cells from the dataset as input, the pathways from NCATS BioPlanet collection with more than 6 overlap genes (1383 pathways) were used as pathway input. UCell (v2.6.2), "score_genes" from Scanpy (v1.9.5), and AUCell (1.24.0) were used for comparison. Moreover, to test the ability of scPAFA to handle large-scale pathway sets, C5 ontology gene sets from MsigDB along with the CRC dataset were applied. After quality control, 14,300 pathways were included for PAS calculation. To ensure consistent computational performance each time the program is executed, we conduct all tests sequentially.

### Comparison of PAS similarity among "fast_ucell", "fast_score_genes", and AUCell

We use 100 thousand cells subset of the CRC dataset (with 1629 pathways) and lupus dataset (with 1383 pathways) to calculate cosine similarity of PAS among "fast_ucell", "fast_score_genes", and AUCell. In brief, for each cell, we generated three PAS vectors containing the PAS of all pathways corresponding to "fast_ucell", "fast_score_genes", and AUCell. The pairwise cosine similarity between these vectors was calculated using the "cosine_similarity" function from scikit-learn (v 1.3.2).

### Application of scPAFA on CRC single-cell dataset

We extracted 108,497 tumor epithelial cells from the CRC dataset AnnData as the input for scPAFA. Python dictionary containing NCATS BioPlanet with 1658 pathways and 3CA metaprogram with 149 gene sets was used as pathway input. After filtering out pathways with fewer than 6 overlap genes with the expression matrix, 1629 pathways remained. PAS matrix was calculated using the "fast_ucell" function, setting max rank ($r_{max}$) as 2000. Firstly, we used single-group MOFA framework to identify potential batch effects, function "generate_scpafa_input" was used for PAS pseudobulk integration with 65 pseudobulk samples (30 MMRp and 35 MMRd), specifying cell type as "view", setting a pseudobulk sample to include at least 10 cells, and a "view" to consist of at least 45 pseudobulk samples. After identifying batch effect in samples' stratification caused by different 10X chemistry versions, we used "generate_scpafa_input_multigroup" to aggregate the PAS matrix for multi-group MOFA+ framework, specifying batch as "group", setting a "view" to consist of more than 75% of pseudobulk samples (at least 15) in each group. In MOFA model training, we set the factor number as 10 and filtered out factors that explained less than 1% of variation. The latent factor matrix and weight matrix from the converged MOFA model were extracted using mofax. We employed "nonparametric_test_category" functions with latent factor matrix and sample-level metadata to identify MMR status-related multicellular pathway modules by the significance of the Mann-Whitney U test. The aforementioned visualization functions are utilized to illustrate the stratification of samples and the biological interpretation of modules.

### Application of scPAFA on the CRC-CRLM dataset

We used the CRC-CRLM dataset with 163,347 cells as the input for scPAFA. As pathway inputs, we utilized NCATS BioPlanet with 1658 pathways and the 3CA metaprogram with 149 gene sets.

After filtering out pathways with fewer than 6 overlapping genes with the expression matrix, 1636 pathways remained. The PAS matrix was then calculated using the fast_ucell function, with the maximum rank ($r_{max}$) set to 2000. Subsequently, we employed single-group MOFA framework; for corresponding pseudobulk integration, we set a pseudobulk sample to include at least 5 cells, and a "view" to consist of at least 20 pseudobulk samples. Similar to the steps described in the CRC dataset, the MOFA model was trained, and multicellular pathway modules that capture the heterogeneity of CRC were identified and interpreted.

## Application of scPAFA on lupus single-cell dataset

The lupus dataset is based on multiplex scRNA-seq with cell-level processing batch information[3]. We extracted 1,067,499 cells from 3 processing batches that simultaneously contain SLE and health controls as scPAFA input, along with 1383 pathways from NCATS BioPlanet which contain more than 6 overlap genes with the expression matrix. We generated the PAS matrix by using the "fast_ucell" function, setting max rank ($r_{max}$) as 2000. Subsequently, we employed single-group MOFA framework to validate the batch effect caused by processing batch; for corresponding pseudobulk integration, we set a pseudobulk sample to include at least 20 cells, and a "view" to consist of at least 220 pseudobulk samples. We observed batch effect in samples' stratification caused by processing batch, we also identified a cluster of 32 pseudobulk samples that may be contaminated by red blood cells by using k-means clustering and scPAFA downstream analysis functions. After filtering out this cluster of samples, the remaining 250 pseudobulk samples were split into a training set with 175 pseudobulk samples (51 healthy controls and 124 SLE) and a testing set with 75 pseudobulk samples (17 healthy controls and 58 SLE). The PAS matrix with 664,214 cells from the training set was aggregated and then used for multi-group MOFA+ framework, specifying processing batch as "group", setting a "view" to consist of more than 75% of pseudobulk samples (at least 15) in each group. Similar to the steps described in the CRC dataset, the MOFA model was trained and SLE-related multicellular pathway modules were identified and interpreted.

## Machine learning model training and evaluation

For the CRC dataset, top-30 pathway-cell type pairs with the highest absolute weights associated with factors 2, 3, and 6 were collected. After removing cell type information and eliminating duplicate values, the PAS of the remained 30 pathways were calculated on GSE39582 and TCGA COAD-READ data using the "gsva" and "ssgsea" functions in gseapy (v1.1.0) and used as features for classifier training. GSE39582 was split into a 70% training set and a 30% test set, whereas TCGA COAD-READ data was used as an independent test set. We used the 'LinearSVC' function with L1 regularization from scikit-learn (v 1.3.2) to train the MMR status classifier. After determining the parameter for L1 regularization by employing 4-fold cross-validation on the training set, we set C as 0.2 and trained the SVC model. The AUROC of the converged model on the test set and independent test set was characterized to evaluate the performance of the model.

For the lupus dataset, the SLE/healthy SVC classifier was trained on scRNA-seq data and bulk RNA data, separately. For scRNA-seq data, PAS of 78 pathway-cell type pairs were extracted from the pseudobulk PAS matrix and used as features for classifier training, which were collected and eliminated duplicates from top-40 pathway-cell type pairs with the highest absolute weights associated with factors 1 and 6. After determining the parameter for L1 regularization by employing 4-fold cross-validation on the training set with 175 pseudobulk samples, we set C as 0.05 and trained the SVC model. The AUROC of the classifier was evaluated on the test set with 75 pseudobulk samples. For bulk RNA data, 78 pathway-cell type pairs were aggregated into 52 pathways and then used as features for classifier training. SLE/healthy SVC classifier was trained and evaluated following a similar process in the abovementioned CRC dataset while using 70% of GSE88884 as the training set, 30% of GSE88884 as the test set, GSE61635 as the independent test set and L1 regularization parameter C as 0.01.

## Comparison of SCPA and scPAFA

To ensure consistency with the input of scPAFA, we applied SCPA (v 1.6.2) on the abovementioned 108,497 tumor epithelial cells from the CRC dataset with 1629 pathways and 664,214 cells training set of the lupus dataset with 1383 pathways. We follow the official tutorial for a systems-level analysis of many cell types in the disease to run SCPA. SCPA supports parallel processing, we used 10 cores for it like the core numbers used in scPAFA. The "compare_pathways" function of SCPA uses a downsampling strategy (selecting 500 cells per condition per cell type as default), we run SCPA by setting downsampling parameters of the "compare_pathways" function to 500, 1000, and 2000 cells, respectively.

In the CRC dataset, the "compare_pathways" function was used to detect the difference activated pathways between MMRd and MMRp patients in each cell type. In the lupus dataset, the "compare_pathways" function was used to detect the difference activated pathways between SLE patients and healthy individuals in each cell type. For SCPA, runtime measurement concludes once the "compare_pathway" function has been executed across all cell types. For scPAFA, it ends after the MOFA model has converged, following the completion of PAS calculation for all cells and pseudobulk integration.

The output of SCPA is a cell type-pathway Qval matrix, where a higher Qval indicates greater differences in the pathway activity between conditions within a cell type. To generate results comparable to this matrix, we combined the weight matrices associated with multiple factors in scPAFA. In CRC dataset, we extracted weight matrix of factors 2, 3, and 6 of all cell types using "get_weights" function of MOFA model, these weights are first converted to their absolute values, the weight of a pathway within a cell type is defined as the maximum value among its weights across factors 2, 3, and 6, thus a cell type-pathway weight matrix was generated. In lupus dataset, similar approach was applied on weight matrix of factors 1 and 6. The cell type-pathway Qval matrix (SCPA) and the cell type-pathway weight matrix (scPAFA) were then filtered to retain only the union of the top 3 pathways in each cell type, and subsequently sorted in descending order based on the mean Qval/weight across all cell types. The filtered matrices are considered to represent the differential pathways highlighted by SCPA and scPAFA.

## Statistics and reproducibility

In our study, the difference of latent factor values between different disease states was examined by the two-sided Mann-Whitney U test (two conditions) as well as two-sided Kruskal-Wallis test (more than 2 conditions), and Benjamini-Hochberg correction for *p*-values was applied. The violin plots showing the difference in cell-level PAS between different disease states were also examined by the Mann-Whitney U test. In the figures, *p*-values or adjusted *p*-values exceeding 0.05 were deemed statistically non-significant and left unlabeled. Values equal to or below 0.05 and 0.01 were denoted with * and **, respectively.

## Reporting summary

Further information on research design is available in the Nature Portfolio Reporting Summary linked to this article.

## Data availability

CRC scRNA-seq data can be accessed at Broad Institute's Single Cell Portal (https://singlecell.broadinstitute.org/single_cell/study/SCP1162). CRC-CRLM dataset can be accessed at https://www.ncbi.nlm.nih.gov/geo/query/acc.cgi?acc=GSE164522. Lupus scRNA-seq data can be accessed at CZ CELLxGENE Annotate(https://cellxgene.cziscience.com/collections/436154da-bcf1-4130-9c8b-120ff9a888f2). GSE39582, GSE61635, and GSE88884 are publicly accessible. TCGA COAD-READ RNA-seq data can be found via the UCSC Xena browser(https://xena.ucsc.edu/). The data[45] for reproducing the analyses and figures presented in this study is available at GitHub (https://github.com/ZhuoliHuang/scPAFA_paper) and Zenodo (https://doi.org/10.5281/zenodo.14039335).

## Code availability

scPAFA is open-source and publicly hosted on GitHub (https://github.com/ZhuoliHuang/scPAFA) and PyPi (https://pypi.org/project/scPAFA/) with documentation and tutorials. The code[45] to reproduce the analyses and figures presented in this study is available at GitHub (https://github.com/ZhuoliHuang/scPAFA_paper) and Zenodo (https://doi.org/10.5281/zenodo.14039335).

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

## Acknowledgements

We acknowledge China National GeneBank (CNGB) for its support of this study.

## Author contributions

Jianhua Yin and Xin Jin led and supervised the study. Zhuoli Huang designed and implemented the scPAFA software, and led the data analyses. Zhuoli Huang, Yuhui Zheng, and Weikai Wang created visualizations of results.

Jianhua Yin, Wenwen Zhou, and Chen Wei provided biological interpretation. Zhuoli Huang, Jianhua Yin, Xiuqing Zhang, and Xin Jin wrote the manuscript. Zhuoli Huang, Yuhui Zheng, Yanbo Zhang, and Jianhua Yin revised the manuscript.

## Competing interests

The authors declare no competing interests.
