## [Transparent Peer Review file · Communications Biology]

Uncovering disease-related multicellular pathway modules on large-scale single-cell transcriptomes with scPAFA

Corresponding Author: Dr Jianhua Yin

Version 0:

Reviewer comments:

Reviewer #1

(Remarks to the Author)

The manuscript presents the scPAFA tool to uncover disease-related multicellular pathway modules on large-scale single-cell RNA-seq data. This tool can efficiently handle large datasets with speed. However, enhancing the methodological detail and clarity of presentation will make the findings and the paper more impactful. Here are my detailed comments and suggestions for revision:

1. The explanation of the methodological framework should be more detailed including the used algorithms and computational strategies, especially in the section multicellular module identification. For example, how to obtain multicellular pathway modules and pathway-cell type pairs?
2. Please provide more detailed information about the computational optimizations that make scPAFA suitable for large-scale data analysis.
3. Please improve the resolution of the figures to enhance readability and effectiveness in conveying the results.
4. Although the author has demonstrated some roles of scPAFA on analyzing cell type-specific pathways in disease, it remains insufficient. As a tool designed for pathway analysis of single-cell RNA-seq data, the author should elaborate on its broader applications, particularly including but not limited in analyzing intra- and inter-patient disease heterogeneity.
5. For CRC datasets, why only analyze batch v2 and v3, where is v1?
6. In the figure 3, 3a showed "METAPROGRAM_6_HYPOXIA" only existed in cell type "Tumor cE02", however it was related to many cell types in 3c, why? Give a reasonable explanation.
7. Some minor errors, such as "...Fig. 1A..." in the line 86; no citations for fig 2f in the line 174; "(Fig. 3a, Supplementary Table 1) Furthermore..." in the line 207; citation error for fig 3f and 3g in the line 233.

Reviewer #2

(Remarks to the Author)

Given the mention of AUCell in the introduction, it would be beneficial for readers if the authors could provide a comparison or discussion regarding how their pathway activity score approach relates to AUCell. Specifically, highlighting the differences, advantages, or potential limitations compared to AUCell could offer valuable insights into the novel contributions of this work.

Reviewer #3

(Remarks to the Author)

Below are some of my concerns and requests for clarification:

How does the performance and scalability of scPAFA change as the size of the gene sets or pathways being analyzed increases? Is scPAFA scalable and able to efficiently handle the analysis of large gene sets or pathways?

The authors mentioned, "Moreover, the value of factor 1 was significantly increased in SLE patients, while factor 6 was significantly decreased (Fig. 4a), hence factors 1 and 6 were identified as SLE-related multicellular pathway modules." However, upon closer inspection, it appears that factors 3, 5, and 7 exhibit a similar pattern of increased values to factor 1 in

SLE patients, while factor 2 displays a comparable decrease in value to factor 6. Given these observations, why did the authors not consider factors 3, 5, 7, and 2 as potential SLE-related multicellular pathway modules alongside factors 1 and 6?

The authors claim that scPAFA is a fast and computationally efficient tool, but they do not provide details on the specific reasons behind its speed and efficiency. Given that scPAFA utilizes the multi-omics factor analysis (MOFA) framework as its base model, is the tool's fast and scalable performance primarily attributed to the use of MOFA, or are there other factors contributing to its computational efficiency?

In the section discussing "MMR Status-Related Multicellular Pathway Modules," the authors mention reformatting the PAS matrix with batch, sample, and cell type information before applying the multi-group MOFA+ framework. However, many single-cell datasets lack comprehensive cell type annotations or may belong to single-sample, single-batch studies. Given these limitations, can scPAFA still produce accurate and reliable multicellular pathway modules if these labels are not available for the input data? Since the converged MOFA model relies on integrating pathway activity differences across cell types, how would the absence of cell type annotations impact scPAFA's ability to identify biologically meaningful multicellular pathway modules?

The authors used the GSVA scores of the high-weight pathways as input features to train a classifier. However, GSVA is a relative scoring method, where the pathway scores depend on the analyzed gene sets. In contrast, methods such as ssGSEA independently score each cell for pathway enrichment without relying on relative comparisons across samples. Given this difference in scoring approaches, can the authors compare the accuracy of their pseudobulk classifier when using ssGSEA scores instead of GSVA scores as input features? Such a comparison could provide insights into the robustness and generalizability of the identified multicellular pathway modules across different enrichment scoring methods.

Can the authors provide a comparative evaluation of scPAFA against the recently proposed SCPA method, particularly in terms of sensitivity, scalability, and biological interpretability on real single-cell datasets? Reference: <https://doi.org/10.1016/j.celrep.2022.111697>.

Version 1:

Reviewer comments:

Reviewer #1

(Remarks to the Author)

The authors have addressed all my concerns.

Reviewer #2

(Remarks to the Author)

Thank you for addressing my concerns. I have reviewed the updated manuscript and am satisfied with the revisions.

Point-by-Point Response to Reviewer

Reviewer #1 (Remarks to the Author):

The manuscript presents the scPAFA tool to uncover disease-related multicellular pathway modules on large-scale single-cell RNA-seq data. This tool can efficiently handle large datasets with speed. However, enhancing the methodological detail and clarity of presentation will make the findings and the paper more impactful.

We sincerely thank you for the suggestions regarding our manuscript. We have modified it according to your suggestions.

Major Comments#1:

1. The explanation of the methodological framework should be more detailed including the used algorithms and computational strategies, especially in the section multicellular module identification. For example, how to obtain multicellular pathway modules and pathway-cell type pairs?

Response #1: We apologize for the unclear explanation of the framework in our previous submission. We have essentially rewritten the 'Overview of scPAFA workflow' section in the Results section and provided more detailed descriptions for each step in the Methods section.

For example, in the current revised version, we have used the following description to explain how to obtain multicellular pathway modules and pathway-cell type pairs:

In the third step, the MOFA model is trained via 'run_mofapy2' function of scPAFA, then the converged model is output. The latent factor matrices are extracted from the converged model using the 'get_factors' function. These matrices are then scaled per factor per group, and subsequently integrated into a single latent factor matrix, with rows representing pseudobulk samples and columns representing factors. Similarly, the feature weight matrices are extracted using the 'get_weights' function, scaled per factor per cell type (view), and integrated into a single weight matrix, with rows representing pathway-cell type pairs and columns representing factors. The factors integrate the primary axes of variation in PAS across conditions from different cell types and are considered as multicellular pathway modules, which can be interpreted by high weight pathway-cell type pairs in the corresponding weight matrix.

For example, in the current revised version, we have used the following description to explain how to identify disease-related multicellular pathway modules (latent factors):

We assumed that multicellular pathway modules (latent factors) with values differing across different disease conditions are related to the disease, to test the significance of differences in factor values between disease states, we provided the 'parametric_test_category' and the 'nonparametric_test_category' functions in scPAFA. These functions automatically detect the number of disease state conditions. If there are two conditions, a Mann-Whitney U test (non-parametric) or t-test (parametric) is used; if there are more than two conditions, a Kruskal-Wallis test (non-parametric) or ANOVA (parametric) is applied. The p-value from these tests was then

adjusted using Benjamini-Hochberg correction across all factors. Factors with adjusted p-values less than 0.05 can be considered related to the disease. Besides, we also provided the 'cal_correlation' function to identify modules associated with continuous traits.

Major Comments#2:

2. Please provide more detailed information about the computational optimizations that make scPAFA suitable for large-scale data analysis.

Response #2: We have provided more detailed information about the computational optimizations in the current revised version.

For example, in the current revised version, we have used the following description to explain the computational optimizations in the Results section:

In brief, the 'fast_ucell' function reimplements the R package UCell in Python language, utilizing more vectorized computations and designing an efficient chunking and concurrent computation process to fully use multi-core CPUs; whereas the 'fast_score_genes' is a concurrent implementation of the single-core 'score_genes' function in Scanpy, designed with chunking capabilities. During the PAS computation, large datasets are first divided into multiple chunks, with each chunk containing 100,000 cells as default. For each pathway, the PAS calculation on a chunk is a fast, vectorized process supported by SciPy and NumPy; to deal with a large number of pathways, the pathways set is partitioned and distributed across multiple cores for parallel computation. This design enables scPAFA to quickly and efficiently convert a cell-gene expression matrix into a cell-pathway PAS matrix (more details in Methods).

For example, in the current revised version, we have used the following description to explain the computational optimizations in the Methods section:

Computational optimizations of 'fast_ucell'

The function 'fast_ucell' is a Python implementation of the R package UCell, which is faster, with similar adjustable parameters, capable of yielding consistent PAS. UCell calculates gene signature scores for scRNA-seq data based on Mann-Whitney U statistic, which is closely related to methods based on AUC scores such as AUCCell. UCell mainly consists of two steps: 1) Given single-cell gene expression matrix X with genes as g and cells as c (e.g. count matrix or log-normalized count matrix), relative gene expression ranks matrix R is calculated by sorting all genes in descending order in each cell. To mitigate the uninformative tail caused by the sparsity of single-cell data, $r_{c,g} = r_{max} + 1$ for all $r_{c,g} > r_{max}$, with $r_{max} = 1500$ by default. 2) For a pathway S composed of n genes (S_1, \dots, S_n), PAS of each cell j in X was calculated with the formula, ranging from 0 to 1:

$$PAS_j = 1 - \frac{U_j}{n * r_{max}}$$

where U_j is the Mann-Whitney U statistic calculated by:

$$U_j = \sum_{i=1}^n r'_{j,i} - \frac{n(n+1)}{2}$$

and R'_j is obtained by sub-setting R_j on the pathway S .

The performance of 'fast_ucell' running on large datasets was enhanced by multi-process parallelism and vectorized computation based on the original UCell computation process. In step 1, single-cell gene expression matrix X in sparse matrix format from large dataset is automatically split into chunks of reduced size (default as 100,000 cells), the relative gene expression ranks matrix R is calculated parallelly by using 'stats.rankdata' function in SciPy (v1.11.3) package, then R is reformatted into sparse matrix by assigning $r_{c,g} = 0$ for all $r_{c,g} > r_{max}$. In step 2, R_{sub} is extracted from R , which consists of the union of genes contained in all pathways in pathway set P , then R_{sub} is split into chunks by cells (default as 100,000 cells) for serial processing while pathway set P is split into chunks by pathways for parallel processing. For a pathway S composed of n genes, the PAS of S in a chunk of R_{sub} is calculated by a vectorized computation process. In brief, R_S is extracted from the chunk of R_{sub} , which only includes all genes in S ; row (cell)-wise summation of the R_S is performed and stored as vector V , and cells with a sum of 0 are labeled as C_0 , while others are labeled as C_1 . For cells in C_0 , the PAS of S is approximately specified as 0. For cells in C_1 , the numbers of zero columns(gene) in R_S are counted as vector z , then PAS of S is calculated with the formula using SciPy package:

$$PAS_{C_1} = 1 - \frac{V_{C_1} + (r_{max} + 1) * z - n(n + 1)/2}{n * r_{max}}$$

To handle a large number of pathways, the pathway set P is divided and distributed across multiple cores for parallel processing using 'ProcessPool' from pathos (v0.3.1).

Computational optimizations of 'fast_score_genes'

The function 'fast_score_genes' is a multiprocessing implementation of the 'score_genes' function in Scanpy ('AddModuleScore' function in Seurat). In function 'score_genes', PAS of a gene signature S in a given cell j of single-cell gene expression matrix X is defined as the average relative expression of the overlap genes of j in S , which can be calculated by a three-step process: 1) using the mean expression across all input cells to categorize all input genes into bins (default set to 25 bins) of uniform size e , 2) randomly choosing reference genes (default as 50 genes) from the same expression bin for each gene in the gene signature S as reference gene set S_{ref} . 3) the PAS of cell j is the average expression of S subtracted with the average expression of reference gene set S_{ref} :

$$PAS_j = \sum_{g \in S} X_{j,g} / n - \sum_{g_{ref} \in S_{ref}} X_{j,g_{ref}} / n_{ref}$$

where $X_{j,g}$ is the expression of gene g in cell j , n and n_{ref} is the number of genes in S and S_{ref} , respectively. The computations in Step 3 are supported by vectorization processes in the SciPy and NumPy (v1.26.1) packages. In 'fast_score_genes', given a single-cell gene expression matrix X , step 1 was performed first, then X can be split into smaller chunks by cells (default as 100,000 cells) for serial processing (minimizing memory usage). For each chunk of X , steps 2 and 3 can be executed concurrently using multiple pathways in pathway set P . Similar to 'fast_ucell', the pathway set P is divided and distributed across multiple cores for parallel processing using 'ProcessPool' from pathos.

Major Comments#3:

3. Please improve the resolution of the figures to enhance readability and effectiveness in conveying the results.

Response #3: Sure, in this revised version, we have provided the original vector graphic files in pdf format of all figures.

Major Comments#4:

4. Although the author has demonstrated some roles of scPAFA on analyzing cell type-specific pathways in disease, it remains insufficient. As a tool designed for pathway analysis of single-cell RNA-seq data, the author should elaborate on its broader applications, particularly including but not limited in analyzing intra- and inter-patient disease heterogeneity.

Response #4: Thank you for your suggestion. We think that MMR status of CRC can be viewed as a form of inter-patient heterogeneity. In the current revised version, we further utilized scPAFA on a new dataset (liu et al) to characterize the differences between CRC tumor and adjacent colon tissues, as well as between primary and metastatic sites, highlighting a form of intra-patient heterogeneity. The corresponding results are shown in the Fig. 4 (new figure) and the Supplementary Fig. 2 (new figure).

Liu et al. "Immune phenotypic linkage between colorectal cancer and liver metastasis." *Cancer cell* 40.4 (2022): 424-437.

Major Comments#5:

5. For CRC datasets, why only analyze batch v2 and v3, where is v1?

Response #5: For CRC dataset (Pelka et al. 2021), there are only two technical batches v2 and v3, which refer to different 10X chemistry versions (Chromium Single Cell 3' Library & Gel Bead Kit v2 and Chromium Single Cell 3' Library & Gel Bead Kit v3), that corresponded to the protocol for generating single-cell gene expression libraries. Pelka et al emphasized this technical batch in their study, which can be found in Table S1 of their research (<https://ars.els-cdn.com/content/image/1-s2.0-S0092867421009454-mmc1.xlsx>).

Currently, we have retained this naming (v2 and v3) in the manuscript to remain consistent with Pelka et al. If you believe it may confuse, we can replace it with 'batch1' and 'batch2' and provide a detailed explanation in the Methods section in the next revision.

Pelka et al. "Spatially organized multicellular immune hubs in human colorectal cancer." *Cell* 184.18 (2021): 4734-4752.

Major Comments#6:

6. In the figure 3, 3a showed "METAPROGRAM_6_HYPOXIA" only existed in cell type "Tumor cE02", however it was related to many cell types in 3c, why? Give a reasonable explanation.

Response #6: In Fig. 3a, due to the limitations of the main figure layout and space, we have only displayed the top 3 positive and negative weight features (pathway-cell type pair) for each factor. Apart from these, many other high-weight features can be used to explain the factor as well, hence we have provided the complete weight matrix as well as the top 30 positive and negative weights of each factor in Supplementary Table 2.

Below is the table of the top 30 negative weight features for Factor 6 in Supplementary Table 2, these features can be considered as MMRd-related pathway-cell type pair:

		weight in Factor6
1	Top30_negative	
2	GAVISH_3CA_MALIGNANT_METAPROGRAM_METAPROGRAM_6_HYPOXIA_View_Tumor cE02 (StemTA-likeImmature Goblet)	-4.217670434
3	bioplanet_955_Glycolysis_View_Tumor cE06 (Immature Goblet)	-3.688657177
4	GAVISH_3CA_METAPROGRAM_EPITHELIAL_ALVEOLAR_View_Tumor cE05 (Enterocyte 2)	-3.688090026
5	bioplanet_50_Lck and Fyn tyrosine kinases in initiation of T cell receptor activation_View_Tumor cE05 (Enterocyte 2)	-3.302086266
6	GAVISH_3CA_METAPROGRAM_B_CELLS_METABOLISM_MYC_View_Tumor cE06 (Immature Goblet)	-3.203054852
7	bioplanet_322_B lymphocyte cell surface molecules_View_Tumor cE05 (Enterocyte 2)	-3.126087715
8	GAVISH_3CA_MALIGNANT_METAPROGRAM_METAPROGRAM_6_HYPOXIA_View_Tumor cE03 (StemTA-like prolif)	-3.038484252
9	GAVISH_3CA_MALIGNANT_METAPROGRAM_METAPROGRAM_6_HYPOXIA_View_Tumor cE01 (StemTA-like)	-3.019062328
10	bioplanet_955_Glycolysis_View_Tumor cE02 (StemTA-likeImmature Goblet)	-2.978115459
11	bioplanet_1276_Proteasome complex_View_Tumor cE06 (Immature Goblet)	-2.920951894
12	bioplanet_874_Degradation of cysteine and homocysteine_View_Tumor cE03 (StemTA-like prolif)	-2.819808001
13	GAVISH_3CA_MALIGNANT_METAPROGRAM_8_PROTEASOMAL_DEGRADATION_View_Tumor cE06 (Immature Goblet)	-2.693458725
14	bioplanet_1743_Mitochondrial fatty acid beta-oxidation_View_Tumor cE05 (Enterocyte 2)	-2.662351234
15	bioplanet_874_Degradation of cysteine and homocysteine_View_Tumor cE01 (StemTA-like)	-2.642935791
16	GAVISH_3CA_METAPROGRAM_B_CELLS_CELL_CYCLE_View_Tumor cE06 (Immature Goblet)	-2.631665716
17	bioplanet_954_Gluconeogenesis_View_Tumor cE06 (Immature Goblet)	-2.627452042
18	GAVISH_3CA_MALIGNANT_METAPROGRAM_METAPROGRAM_6_HYPOXIA_View_Tumor cE06 (Immature Goblet)	-2.601065002
19	bioplanet_256_SREBP control of lipid biosynthesis_View_Tumor cE01 (StemTA-like)	-2.591320968
20	GAVISH_3CA_METAPROGRAM_FIBROBLASTS_MHC_II_CYTOKINE_View_Tumor cE05 (Enterocyte 2)	-2.585149699
21	bioplanet_1482_Mitochondrial beta-oxidation of saturated fatty acids_View_Tumor cE01 (StemTA-like)	-2.573870734
22	bioplanet_321_Bystander B cell activation_View_Tumor cE05 (Enterocyte 2)	-2.535853814
23	bioplanet_211_Eosinophils in the chemokine network of allergy_View_Tumor cE05 (Enterocyte 2)	-2.524164999
24	bioplanet_464_Antigen processing and presentation_View_Tumor cE05 (Enterocyte 2)	-2.519492547
25	bioplanet_1482_Mitochondrial beta-oxidation of saturated fatty acids_View_Tumor cE03 (StemTA-like prolif)	-2.510535409
26	bioplanet_332_Alternative complement pathway_View_Tumor cE02 (StemTA-likeImmature Goblet)	-2.458841832
27	GAVISH_3CA_METAPROGRAM_MACROPHAGES_MES_GLYCOLYSIS_View_Tumor cE01 (StemTA-like)	-2.347989104
28	GAVISH_3CA_METAPROGRAM_MACROPHAGES_MES_GLYCOLYSIS_View_Tumor cE02 (StemTA-likeImmature Goblet)	-2.342451503
29	bioplanet_846_Aflatoxin B1 metabolism_View_Tumor cE03 (StemTA-like prolif)	-2.33922896
30	bioplanet_829_Pyruvate metabolism and citric acid (TCA) cycle_View_Tumor cE05 (Enterocyte 2)	-2.321573248
31	bioplanet_1408_Licensing factor removal from origins_View_Tumor cE06 (Immature Goblet)	-2.319622318

A Part of Supplementary Table 2

We can find that “METAPROGRAM_6_HYPOXIA” in Tumor cE02, Tumor cE03, Tumor cE01 and Tumor cE06 is the top1, top7, top8, and top17 features related to MMRd, respectively. Correspondingly, In the figure 3c, the cell-level PAS of “METAPROGRAM_6_HYPOXIA” is significantly higher in MMRd than MMRp in Tumor cE01, Tumor cE02, Tumor cE03 and Tumor cE06, respectively.

Major Comments#7:

7. Some minor errors, such as “...Fig. 1A...” in the line 86; no citations for fig 2f in the line 174; “(Fig. 3a, Supplementary Table 1) Furthermore...” in the line 207; citation error for fig 3f and 3g in the line 233.

Response #7: Thank you for your thorough review. We have corrected these errors in the current revised version.

Reviewer #2 (Remarks to the Author):

Major Comments#1:

Given the mention of AUCell in the introduction, it would be beneficial for readers if the authors could provide a comparison or discussion regarding how their pathway activity score approach relates to AUCell. Specifically, highlighting the differences, advantages, or potential limitations compared to AUCell could offer valuable insights into the novel contributions of this work.

Response #1: Thank you for your suggestion, we fully agree with your perspective. In the current revised version, we have provided a comparison between the AUCell method and pathway activity score approach ('fast_uccell' and 'fast_score_genes') used in scPAFA in the Results and Discussion sections.

It was observed that the running speeds of functions 'fast_uccell' (10 cores) and 'fast_score_genes' (10 cores) were approximately 4.4 to 11.4 times faster than AUCell method (10 cores) on the dataset range from 10,000 cells to 1,263,676 cells. For instance, on the complete lupus dataset, AUCell cost 5.1 hours; whereas 'fast_uccell' cost 27.05 minutes and 'fast_score_genes' cost 29.9 minutes (Fig. 1b and Supplementary Table 1).

Fig. 1b

The 'fast_uccell' (In Python) is a more computationally efficient implementation of UCell (R package), which can provide the same PAS of UCell. UCell calculates PAS based on the Mann-Whitney U statistic, and AUCell uses the area under the curve (AUC) of the recovery curve. They both rely on the relative ranking of genes within individual cells, making their PAS robust to variations in dataset composition. We found that 'fast_uccell' PAS was closely similar to AUCell PAS by setting the same max rank threshold, the average cosine similarity between 'fast_uccell' PAS and AUCell PAS was 0.97 and 0.94 in the CRC dataset and lupus dataset, respectively (Fig. 1d and Supplementary Table 1).

Fig. 1d

Furthermore, 'fast_score_genes' normalizes its PAS against the average expression of a control set of genes across the entire dataset, making it sensitive to the composition. Consistently, the PAS of 'fast_score_genes' has a moderate degree of similarity with the 'fast_uccell' PAS or AUCell PAS (Fig. 1d and Supplementary Table 1).

Reviewer #3 (Remarks to the Author):

Major Comments#1:

How does the performance and scalability of scPAFA change as the size of the gene sets or pathways being analyzed increases? Is scPAFA scalable and able to efficiently handle the analysis of large gene sets or pathways?

Response #1: Thank you for your question. In the current revised version, we tested the efficiency of the 'fast_ucell' and 'fast_score_genes' functions of scPAFA in handling a large number (> 10,000) of pathways. The corresponding results are shown in Fig. 1c. We used C5 ontology gene sets from MsigDB along with the CRC dataset (371,223 cells) for PAS calculation. After quality control, 14,300 pathways were included. Using 20 cores, We found that 'fast_ucell' cost 43.96 minutes and 'fast_score_genes' cost 114.05 minutes, indicating 'fast_ucell' is more efficient in this context, whereas 'fast_score_genes' cost less memory.

Fig. 1c

Major Comments#2:

The authors mentioned, "Moreover, the value of factor 1 was significantly increased in SLE patients, while factor 6 was significantly decreased (Fig. 4a), hence factors 1 and 6 were identified as SLE-related multicellular pathway modules." However, upon closer inspection, it appears that factors 3, 5, and 7 exhibit a similar pattern of increased values to factor 1 in SLE patients, while factor 2 displays a comparable decrease in value to factor 6. Given these observations, why did the authors not consider factors 3, 5, 7, and 2 as potential SLE-related multicellular pathway modules alongside factors 1 and 6?

Response #2: We assumed that multicellular pathway modules (latent factors) with values differing across different disease conditions are related to the disease, to test the significance of differences in factor values between disease states, we provided the 'parametric_test_category' and the 'nonparametric_test_category' functions in scPAFA. These functions automatically detect the number of disease state conditions. If there are two conditions, a Mann-Whitney U test (non-parametric) or t-test (parametric) is used; if there are more than two conditions, a Kruskal-Wallis test (non-parametric) or ANOVA (parametric) is applied. The p-value from these tests was then adjusted using Benjamini-Hochberg correction across all factors. Factors with adjusted p-values less than 0.05 can be considered related to the disease.

We used 'nonparametric_test_category' to test the significance in our study, for the lupus dataset, only the difference in the values of factors 1 and 6 between SLE patients and healthy

individuals is significant with an adjusted p-value of less than 0.05:

Factor	p_value	p_adj	method
Factor1	9.79E-13	7.83E-12	Mann-Whitney U test
Factor2	2.79E-02	7.45E-02	Mann-Whitney U test
Factor3	1.27E-01	1.70E-01	Mann-Whitney U test
Factor4	2.15E-01	2.46E-01	Mann-Whitney U test
Factor5	5.54E-02	1.11E-01	Mann-Whitney U test
Factor6	8.66E-05	3.46E-04	Mann-Whitney U test
Factor7	1.08E-01	1.70E-01	Mann-Whitney U test
Factor8	8.86E-01	8.86E-01	Mann-Whitney U test

The table shows the adjusted p-value of factors in the lupus dataset. Additionally, in Fig. 5a (formerly Fig. 4a), Factor 1 and Factor 6 are marked with "**," indicating that the adjusted p-value is less than 0.01.

Major Comments#3:

The authors claim that scPAFA is a fast and computationally efficient tool, but they do not provide details on the specific reasons behind its speed and efficiency. Given that scPAFA utilizes the multi-omics factor analysis (MOFA) framework as its base model, is the tool's fast and scalable performance primarily attributed to the use of MOFA, or are there other factors contributing to its computational efficiency?

Response #3: The scPAFA framework primarily involves the calculation of PAS (convert a cell-gene expression matrix into a cell-pathway PAS matrix) and the application of the MOFA framework. We claim that scPAFA is a fast and computationally efficient tool primarily because our PAS calculation step ('fast_ucell' and 'fast_score_genes' functions) is much faster than existing methods including UCell, AUCell and 'score_genes' in Scanpy (Fig. 1), especially in large-scale scRNA-seq data, which is contributed by our optimizations in programming of 'fast_ucell' and 'fast_score_genes' functions. For instance, on the complete lupus dataset (1,263,676 cells), UCell cost 21.4 hours for 1383 pathways, 'score_genes' cost 9.3 hours, AUCell cost 5.1 hours; whereas 'fast_ucell' cost 27.05 minutes and 'fast_score_genes' cost 29.9 minutes (Fig. 1b, Supplementary Table 1). In addition, PAS calculation and MOFA framework application are independent, which means that scPAFA can be used solely for the rapid calculation of PAS. About the application of the MOFA framework in scPAFA, in each group and view, cell-level PAS is aggregated into pseudobulk-level PAS across samples/donors by computing the arithmetic mean. Consequently, the input to the MOFA model comprises pseudobulk-level PAS rather than cell-level PAS, enabling the model to be trained efficiently, typically within a few seconds. At the scale of the entire scPAFA framework, the main speed-limiting step is the PAS calculation. Therefore, the optimization we performed in this step makes scPAFA a fast and computationally efficient tool, which is suitable for large-scale scRNA-seq data analysis.

To avoid misunderstanding, we have provided more detailed information about the computational optimizations in the current revised version.

For example, in the current revised version, we have used the following description to explain the computational optimizations in the Results section:

In brief, the 'fast_ucell' function reimplements the R package UCell in Python language, utilizing more vectorized computations and designing an efficient chunking and concurrent computation process to fully use multi-core CPUs; whereas the 'fast_score_genes' is a concurrent implementation of the single-core 'score_genes' function in Scanpy, designed with chunking

capabilities. During the PAS computation, large datasets are first divided into multiple chunks, with each chunk containing 100,000 cells as default. For each pathway, the PAS calculation on a chunk is a fast, vectorized process supported by SciPy and NumPy; to deal with a large number of pathways, the pathways set is partitioned and distributed across multiple cores for parallel computation. This design enables scPAFA to quickly and efficiently convert a cell-gene expression matrix into a cell-pathway PAS matrix (more details in Methods).

For example, in the current revised version, we have used the following description to explain the computational optimizations in the Methods section:

Computational optimizations of ‘fast_ucell’

The function ‘fast_ucell’ is a Python implementation of the R package UCell, which is faster, with similar adjustable parameters, capable of yielding consistent PAS. UCell calculates gene signature scores for scRNA-seq data based on Mann-Whitney U statistic, which is closely related to methods based on AUC scores such as AUCCell. UCell mainly consists of two steps: 1) Given single-cell gene expression matrix X with genes as g and cells as c (e.g. count matrix or log-normalized count matrix), relative gene expression ranks matrix R is calculated by sorting all genes in descending order in each cell. To mitigate the uninformative tail caused by the sparsity of single-cell data, $r_{c,g} = r_{max} + 1$ for all $r_{c,g} > r_{max}$, with $r_{max} = 1500$ by default. 2) For a pathway S composed of n genes (S_1, \dots, S_n), PAS of each cell j in X was calculated with the formula, ranging from 0 to 1:

$$PAS_j = 1 - \frac{U_j}{n * r_{max}}$$

where U_j is the Mann-Whitney U statistic calculated by:

$$U_j = \sum_{i=1}^n r'_{j,i} - \frac{n(n+1)}{2}$$

and R'_j is obtained by sub-setting R_j on the pathway S .

The performance of ‘fast_ucell’ running on large datasets was enhanced by multi-process parallelism and vectorized computation based on the original UCell computation process. In step 1, single-cell gene expression matrix X in sparse matrix format from large dataset is automatically split into chunks of reduced size (default as 100,000 cells), the relative gene expression ranks matrix R is calculated parallelly by using ‘stats.rankdata’ function in SciPy (v1.11.3) package, then R is reformatted into sparse matrix by assigning $r_{c,g} = 0$ for all $r_{c,g} > r_{max}$. In step 2, R_{sub} is extracted from R , which consists of the union of genes contained in all pathways in pathway set P , then R_{sub} is split into chunks by cells (default as 100,000 cells) for serial processing while pathway set P is split into chunks by pathways for parallel processing. For a pathway S composed of n genes, the PAS of S in a chunk of R_{sub} is calculated by a vectorized computation process. In brief, R_S is extracted from the chunk of R_{sub} , which only includes all genes in S ; row (cell)-wise summation of the R_S is performed and stored as vector V , and cells with a sum of 0 are labeled as C_0 , while others are labeled as C_1 . For cells in C_0 , the PAS of S is approximately specified as 0. For cells in C_1 , the numbers of zero columns(gene) in R_S are counted as vector z , then PAS of S is calculated with the formula using SciPy package:

$$PAS_{C_1} = 1 - \frac{V_{C_1} + (r_{max} + 1) * z - n(n + 1)/2}{n * r_{max}}$$

To handle a large number of pathways, the pathway set P is divided and distributed across multiple cores for parallel processing using 'ProcessPool' from pathos (v0.3.1).

Computational optimizations of 'fast_score_genes'

The function 'fast_score_genes' is a multiprocessing implementation of the 'score_genes' function in Scanpy ('AddModuleScore' function in Seurat). In function 'score_genes', PAS of a gene signature S in a given cell j of single-cell gene expression matrix X is defined as the average relative expression of the overlap genes of j in S , which can be calculated by a three-step process: 1) using the mean expression across all input cells to categorize all input genes into bins (default set to 25 bins) of uniform size e , 2) randomly choosing reference genes (default as 50 genes) from the same expression bin for each gene in the gene signature S as reference gene set S_{ref} . 3) the PAS of cell j is the average expression of S subtracted with the average expression of reference gene set S_{ref} :

$$PAS_j = \sum_{g \in S} X_{j,g} / n - \sum_{g_{ref} \in S_{ref}} X_{j,g_{ref}} / n_{ref}$$

where $X_{j,g}$ is the expression of gene g in cell j , n and n_{ref} is the number of genes in S and S_{ref} , respectively. The computations in Step 3 are supported by vectorization processes in the SciPy and NumPy (v1.26.1) packages. In 'fast_score_genes', given a single-cell gene expression matrix X , step 1 was performed first, then X can be split into smaller chunks by cells (default as 100,000 cells) for serial processing (minimizing memory usage). For each chunk of X , steps 2 and 3 can be executed concurrently using multiple pathways in pathway set P . Similar to 'fast_ucll', the pathway set P is divided and distributed across multiple cores for parallel processing using 'ProcessPool' from pathos.

Major Comments#4:

In the section discussing "MMR Status-Related Multicellular Pathway Modules," the authors mention reformatting the PAS matrix with batch, sample, and cell type information before applying the multi-group MOFA+ framework. However, many single-cell datasets lack comprehensive cell type annotations or may belong to single-sample, single-batch studies. Given these limitations, can scPAFA still produce accurate and reliable multicellular pathway modules if these labels are not available for the input data? Since the converged MOFA model relies on integrating pathway activity differences across cell types, how would the absence of cell type annotations impact scPAFA's ability to identify biologically meaningful multicellular pathway modules?

Response #4: Thank you for your question, we are pleased to have the opportunity to discuss some of the features and limitations of scPAFA in detail:

- (1) If scPAFA is used solely for the rapid calculation of cell-level PAS, it only requires an expression matrix and a pathway list as inputs.
- (2) If scPAFA is used for producing multicellular pathway modules, cell-level metadata, including cell type annotation and sample information, is mandatory. This is because, in each cell type, cell-level PAS is aggregated into pseudobulk-level PAS across samples by computing the arithmetic mean, pseudobulk-level PAS is then used as the input to train the MOFA model. Therefore, cell type annotation and sample information are necessary. The scPAFA cannot identify multicellular pathway modules in the absence of cell type annotations.

- (3) The scPAFA cannot identify multicellular pathway modules in single-sample datasets because the MOFA model requires input from more than 15 samples (<https://biofam.github.io/MOFA2/faq.html>).
- (4) Single-batch dataset (with more than 15 samples) is available for scPAFA. Batch information is not mandatory. Given clear batch effects, we can designate batches as 'group' in multi-groups MOFA+ framework to identify shared effects between batches. If batch effects do not exist, we can use the single-group MOFA framework for inference. The scPAFA provides the 'generate_scpafo_input_multigroup' function for applying the multi-group MOFA+ framework and the 'generate_scpafo_input' function for applying the single-group MOFA framework.

To clarify the intended use of scPAFA, we position it as a tool for downstream biological interpretation, particularly well-suited for large-scale scRNA-seq studies involving numerous cells and many samples. Given that a typical primary step in scRNA-seq data analysis is to partition the cells into clusters and annotate them as cell types, we assume that users have already completed cell type annotation before conducting scPAFA analysis.

Major Comments#5:

The authors used the GSVA scores of the high-weight pathways as input features to train a classifier. However, GSVA is a relative scoring method, where the pathway scores depend on the analyzed gene sets. In contrast, methods such as ssGSEA independently score each cell for pathway enrichment without relying on relative comparisons across samples. Given this difference in scoring approaches, can the authors compare the accuracy of their pseudobulk classifier when using ssGSEA scores instead of GSVA scores as input features? Such a comparison could provide insights into the robustness and generalizability of the identified multicellular pathway modules across different enrichment scoring methods.

Response #5: Thank you for your suggestion. We have trained the classifier base on ssGSEA scores in the current revised version. The corresponding results are shown in Supplementary Fig. 1 c-f and Supplementary Fig. 3 e-h. In brief, the accuracy of the classifier based on ssGSEA scores is similar to that of the classifier based on GSVA scores. For the CRC dataset, the AUROC of the classifier based on ssGSEA scores is 0.91 on the test set and 0.95 on the independent test set (Supplementary Fig. 1 d,e), whereas the AUROC of the classifier based on GSVA scores is 0.91 on both the test set and the independent test set (Fig. 3f). For the lupus dataset, the AUROC of the classifier based on ssGSEA scores is 0.94 on both bulk test set 1 and bulk test set 2 (Supplementary Fig. 1 f,g), while the AUROC of the classifier based on GSVA scores is 0.93 on bulk test set 1 and 0.98 on bulk test set 2 (Fig. 6k).

Major Comments#6:

Can the authors provide a comparative evaluation of scPAFA against the recently proposed SCPA method, particularly in terms of sensitivity, scalability, and biological interpretability on real single-cell datasets? Reference: (<https://doi.org/10.1016/j.celrep.2022.111697>).

Response #6: Thank you for your suggestion. Sure, we have provided a comparison of SCPA and

scPAFA on real single-cell dataset in the current revised version. The corresponding results are shown in Fig. 7 (new figure) and the Supplementary Fig. 4 (new figure).

SCPA is a specialized tool for pathway analysis in scRNA-seq data that evaluates the multivariate distributions of pathways to determine their significant difference across conditions. Instead of calculating PAS, it provides a Qval that reflects the variation in pathway activity. SCPA can find the variation of pathway activity across conditions within each cell type, and provide a cell type-pathway Qval matrix. We compared the computational efficiency and the ability to detect pathway activity differences between SCPA and scPAFA on CRC and lupus dataset. To ensure consistency with the input of scPAFA, we applied SCPA (v1.6.2) on the 108,497 tumor epithelial cells from the CRC dataset with 1629 pathways and 664,214 cells training set of the lupus dataset with 1383 pathways. We follow the official tutorial for a systems-level analysis of many cell types in the disease to run SCPA (https://jackbibby1.github.io/SCPA/articles/disease_comparison.html). SCPA supports parallel processing, we used 10 cores for it like the core numbers used in scPAFA. The 'compare_pathways' function of SCPA uses a downsampling strategy (selecting 500 cells per condition per cell type as default), we run SCPA by setting downsampling parameters of the 'compare_pathways' function to 500, 1000, and 2000 cells, respectively.

In the CRC dataset, the 'compare_pathways' function was used to detect the difference activated pathways between MMRd and MMRp patients in each cell type. In the lupus dataset, the 'compare_pathways' function was used to detect the difference activated pathways between SLE patients and healthy individuals in each cell type. For SCPA, runtime measurement concludes once the "compare_pathway" function has been executed across all cell types. For scPAFA, it ends after the MOFA model has converged, following the completion of PAS calculation for all cells and pseudobulk integration.

The output of SCPA is a cell type-pathway Qval matrix, where a higher Qval indicates greater differences in the pathway activity between conditions within a cell type. To generate results comparable to this matrix, we combined the weight matrices associated with multiple factors in scPAFA. In CRC dataset, we extracted the weight matrix of factors 2, 3, and 6 of all cell types using the 'get_weights' function of the MOFA model, these weights are first converted to their absolute values, the weight of a pathway within a cell type is defined as the maximum value among its weights across factors 2, 3, and 6, thus a cell type-pathway weight matrix was generated. In lupus dataset, similar approach was applied on the weight matrix of factors 1 and 6. The cell type-pathway Qval matrix (SCPA) and the cell type-pathway weight matrix (scPAFA) were then filtered to retain only the union of the top 3 pathways in each cell type, and subsequently sorted in descending order based on the mean Qval/weight across all cell types. The filtered matrices are considered to represent the differential pathways highlighted by SCPA and scPAFA.

We observed that as the number of downsampled cells increased, the computation time for SCPA grew dramatically (Fig. 7a, b). In contrast, scPAFA was approximately 1.3(500 cells in lupus dataset) to 103.9 (2000 cells in CRC dataset) times faster than SCPA, even when scPAFA calculated PAS for all cells (Fig. 7a, b and Supplementary Table 1).

In CRC dataset, scPAFA emphasizes the most notable differential pathway between MMRp and MMRd is 'GAVISH_3CA_MALIGNANT_METAPROGRAM_17_INTERFERON_MHC_II_1'(Fig. 7c),

which was consistent with the gene program reported in the original study of Pelka et al. However, this pathway was not emphasized in the SCPA result (Fig. 7c and Supplementary Fig. 4a). It was widely reported that increased type 1 interferon (interferon-alpha/beta) signaling is the hallmark of SLE. Consistently, in lupus dataset, scPAFA emphasizes the most notable differential pathway is 'bioplanet_686_Interferon alpha/beta signaling', which was also detected by SCPA (2000 cells) (Fig. 7d). However, this pathway was not emphasized in the result of was SCPA (500 cells) (Supplementary Fig. 4a), indicating downsampling strategy of SCPA can result in the loss of information in large datasets.

In brief, compared with SCPA, scPAFA can more efficiently utilize data from all cells, avoiding potential information loss and thereby detecting reliable differential pathways in large-scale datasets. Additionally, scPAFA can provide modules that integrate differential information across multiple cell types. However, scPAFA also has limitations. Most importantly, scPAFA requires studies to include at least 15 samples (not cells), which is the requirement of using the MOFA model for identifying multicellular pathway modules. In studies with samples less than 15, SCPA is more flexible and applicable than scPAFA. It is worth mentioning that scPAFA can also be used solely for the rapid calculation of PAS, in which case the sample size does not impact the results.

Fig. 7

Supplementary Fig. 4

	CRC_dataset_time(minutes)	CRC_dataset_mem(GB)		lupus_dataset_time(minutes)	lupus_dataset_mem (GB)
scPA_500	17.08	49.95		19.87	101.6
scPA_1000	103.6	57.69		104.71	106.4
scPA_2000	822.65	73.71		792	120.89
scPAFA_allcells	7.92	57.07		15.36	53.36

A part of Supplementary Table 1

Pelka et al. "Spatially organized multicellular immune hubs in human colorectal cancer." *Cell* 184.18 (2021): 4734-4752.